# SHOULD BIAS BE ELIMINATED? A GENERAL FRAMEWORK TO USE BIAS FOR OOD GENERALIZATION

## ABSTRACT

Most approaches to out-of-distribution (OOD) generalization learn domain-invariant representations by discarding contextual bias. In this paper, we raise a critical question: Should bias be eliminated? If not, is there a general way to leverage it for better OOD generalization? To answer these questions, we first provide a theoretical analysis that characterizes the circumstances in which biased features contribute positively. Although theoretical results show that bias may sometimes play a positive role, leveraging it effectively is non-trivial, since its harmful and beneficial components are often entangled. Recent advances have sought to refine the prediction of bias by presuming reliable prediction from invariant features. However, such assumptions may be too strong in the real world, especially when the target also shifts from training to testing domains. Motivated by this challenge, we introduce a framework to leverage bias in a more general scenario. Specifically, we employ a generative model to capture the data generation process and identify the underlying bias factors, which are then used to construct a bias-aware predictor. Since the bias-aware predictor may shift across environments, we first estimate the environment state to train predictors under different environments, combining them as a mixture of domain experts for the final prediction. Then, we built a general invariant predictor, which can be invariant under label shift, to guide the adaptation of the bias-aware predictor. Evaluations on synthetic data and standard domain generalization benchmarks demonstrate that our method consistently outperforms both invariance-only baselines and recent bias-utilization approaches, yielding improved robustness and adaptability.

## 1 INTRODUCTION

A widely used example of data bias is the "cow vs. camel classification" problem. As illustrated in Figure 1(a), models trained on imbalanced datasets may rely on biased background features for prediction. Specifically, cows are typically photographed in grassy fields, while camels appear in desert settings. Consequently, the model may mistakenly learn to associate the background context, rather than the animal itself, with the class label. From a causal perspective, this issue arises from spurious correlations, where the background acts as a confounding variable influencing both the images and the associated label, as shown in Figure 1(b).

Most OOD generalization methods, such as IRM (Arjovsky et al., 2019), DANN (Ganin et al., 2016a), and Self-Training (Lee et al., 2013), regard data bias as the primary obstacle and therefore aim to learn domain-invariant representations (see Appendix A for further discussion). The core assumption of these methods is that biased features are uninformative, and only invariant features should be transferred. As illustrated in Figure 1(c), when an image is ambiguous (e.g., cow vs. camel), humans often have to rely on background context to make an educated guess. This intuitively suggests that, rather than being entirely detrimental, in some scenarios, the bias derived from background information can sometimes serve as a useful reference in the prediction process.

To analyze the effect of bias, we provide a theoretical framework that characterizes when data bias is identifiable and beneficial for prediction. Our results show that, in general, biased features can be exploited when they can be disentangled from invariant content, and the dependence between biased features and labels is not screened off by invariant content. It indicates that we can disentangle the bias component and study its predictive contribution, enabling bias features to provide complementary information that can be effectively exploited at inference.

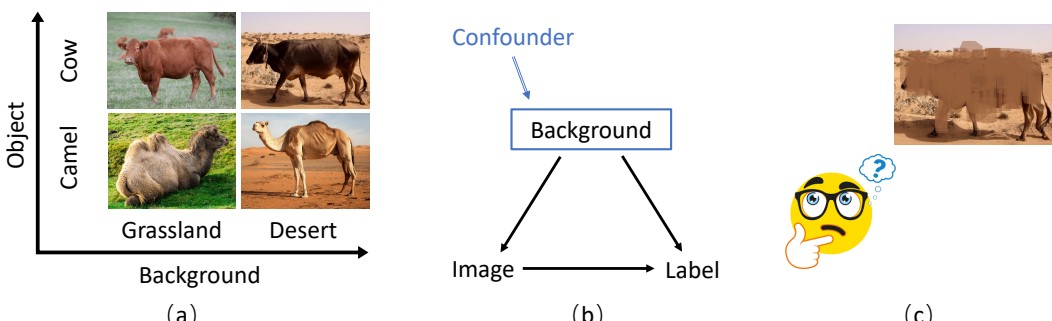

Figure 1: **Illustration of data bias in the "Cow vs. Camel Classification" problem.** (a) An example of the Cow vs. Camel classification task, where both cows and camels are observed in distinct backgrounds, such as grasslands and deserts. (b) A causal graph illustrating spurious correlations introduced by confounders. (c) An intuitive example demonstrating how humans classify an image with ambiguous content as either a cow or a camel.

Although bias contains useful information, using it for prediction is challenging. The most relevant prior work that leverages bias for OOD generalization is Stable Feature Boosting (SFB) (Eastwood et al., 2024). SFB splits the representation into invariant and biased components, using the predictions based on the invariant features as pseudo-labels for test-time adaptation of the biased component. However, this method relies on the assumption that invariant features alone can yield accurate and robust predictions. In particular, it presumes that (1) invariant features contain sufficient information to support prediction, and (2) the prediction target itself remains invariant, i.e., no label shift (Lipton et al., 2018) occurs. In real-world applications, the invariant content may be under-learned due to limited domain heterogeneity, insufficient supervision, or an intrinsic lack of information. Meanwhile, the prediction using the invariant feature may shift in the testing domain, as shown in the common label shift setting (Li et al., 2023b; Wu et al., 2021; Koh et al., 2020).

To leverage bias in more general and realistic settings, we adopt a generative model that factorizes representations into invariant content and contextual bias in an identifiable manner. Building on the identified bias features, we uncover two pathways through which bias can contribute to prediction. Firstly, we use the bias features to estimate environment states and design an environment routing mechanism that leverages these states to gate a mixture of predictors, each conditioned on a specific environment. This environment estimation enables the model to exploit bias as a useful signal, particularly in cases where the invariant content alone is insufficient for accurate prediction. Secondly, we extend the bias correction in SFB (Eastwood et al., 2024) with an adaptive label prior. We compose predictors using bias and content together by Bayes' Rule into two parts: bias predictor and invariant predictor which consists of content predictor and label prior. In addition to correcting the bias predictor using content predictions as pseudo-labels with a fixed label prior, we further learn the label prior in the training stage. This modification can effectively address the problem when both label shift and covariate shift co-exist. The main contributions of this paper are summarized as follows:

- We revisit the role of bias in OOD generalization, highlighting circumstances in which bias offers a constructive signal rather than a nuisance to be eliminated.
- We provide a theoretical analysis that specifies identification and utility conditions for data bias, including cases with concurrent shifts in input and label distributions.
- We introduce a general bias-aware framework that make predictions with an environment routing mechanism and conducts adaptive bias correction.
- We evaluate the proposed method on synthetic and real-world datasets and show that it consistently outperforms baseline approaches across both settings.

## 2 THEORETICAL ANALYSIS

### 2.1 PROBLEM SETUP AND DATA GENERATION PROCESS

**Notation and problem setup.** We formulate the task as a multi-source domain generalization problem, aiming to learn a predictor that remains robust in an unseen target domain using only

data from multiple source domains. Let $\mathbf{x}_k$ denote a high-dimensional image observation $\mathbf{x}_k :=$ $[x_1, \cdots, x_{n_x}] \in \mathcal{X} \subset \mathbb{R}^{n_x}$, and $y_k$ denote the corresponding label. With access to $M$ source domains $\{\mathcal{S}_1, \mathcal{S}_2, ..., \mathcal{S}_M\}$, with each domain represented as $(\mathbf{x}^{\mathcal{S}_i}, \mathbf{y}^{\mathcal{S}_i}) = (\mathbf{x}_k^{\mathcal{S}_i}, y_k^{\mathcal{S}_i})_{k=1}^{m_i}$, our goal is to learn a predictor $f(\mathbf{x})$ that can generalize well to the new domain $\mathcal{T}$ where $\mathcal{T} \notin \{\mathcal{S}_1, \mathcal{S}_2, ..., \mathcal{S}_M\}$.

**Data generation process.** To illustrate the origin and impact of bias, we model the data generation process using a latent variable model. Let $\mathbf{c} := [c_1, \cdots, c_{n_c}] \in \mathcal{C} \subset \mathbb{R}^{n_c}$ denote content variables and $\mathbf{b} := [b_1, \cdots, b_{n_b}] \in \mathcal{B} \subset \mathbb{R}^{n_b}$ denote the bias. As shown in Figure 2, the generation process of the observation $\mathbf{x}$ is defined as $\mathbf{x} := g(\mathbf{c}, \mathbf{b})$, where $\mathbf{z} := [\mathbf{c}, \mathbf{b}]$ with $n_z = n_c + n_b$ dimension denotes the latent variables and $g : [\mathbf{c}, \mathbf{b}] \mapsto \mathbf{x}$ denotes the generating function. To distinguish $\mathbf{c}$ and $\mathbf{b}$, we introduce an environment variable $e$, where the content $\mathbf{c}$ remains invariant despite changes in the environment once $y$ is given, whereas the bias $\mathbf{b}$ varies as the environment changes. Taking the "cow vs. camel classification" problem for illustration, $\mathbf{c}$ denotes the characteristics of the object, like the color/shape of the animal, while $\mathbf{b}$ represents the background information. Note that we consider a more general case where $y$ may also be influenced by the environment, allowing for a more complex relationship modeling between the data and the environment. This assumption is reasonable, as the probability of encountering a cow versus a camel is strongly influenced by the environment (e.g., grasslands vs. deserts) in real-world scenarios. Here we highlight two causal paths between bias and label target, $y \leftarrow e \rightarrow \mathbf{b}$ and $y \rightarrow \mathbf{b}$, which motivate two ways to leverage the bias for prediction. First, we can estimate the environment state from $\mathbf{b}$ and use such information

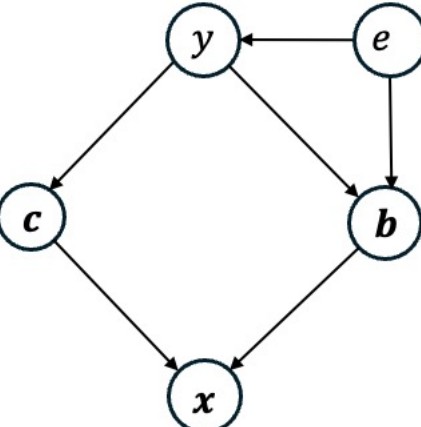

Figure 2: **A graphical representation of the data generation process.** The content variables $\mathbf{c}$ are invariant to environment changes once label $y$ is given, while the bias variables $\mathbf{b}$ vary across environments. Observed data $\mathbf{x}$ is generated by $g(\mathbf{c}, \mathbf{b})$, with $\mathbf{c}$ and $\mathbf{b}$ forming the latent variable $\mathbf{z}$. The environment $e$ affects $\mathbf{b}$ but not $\mathbf{c}$, and the target $y$ may also depend on $e$, reflecting complex data-environment interactions.

to adjust the prediction. Second, we can leverage content to guide the direct prediction from $\mathbf{b}$ to $y$.

## 2.2 THEORETICAL CONDITIONS FOR IDENTIFIABILITY AND USAGE

**Bias and Content identification.** We demonstrate that content and bias variables can be effectively disentangled with an identifiability guarantee. We extend the theoretical results of the previous causal representation learning methods to align with our formulation, with the detailed modification and proof that can be found in Appendix B.1. Such block-wise identification ensures that the learned content and bias representations, $\hat{\mathbf{c}}$ and $\hat{\mathbf{b}}$, are functions of the ground truth content $\mathbf{c}$ and bias $\mathbf{b}$, respectively. It is important to note that this identifiability result has already been established in previous work and is not claimed as a contribution of this paper. However, despite the identification results, existing methods still rely on invariant prediction (including invariant content variable and invariant relation between bias variable and label). Unlikely, building on this identifiability framework, we go further by exploring how to effectively utilize biased data to enhance OOD performance.

**Lemma 2.1.** *(**Block-wise Identification of $\mathbf{c}$ and $\mathbf{b}$** (Kong et al., 2023)). Assuming that the data generation process follows Figure 2, and the following assumptions hold true:*

- *A1 (Smooth and positive density): $p_{\mathbf{z}|e,\mathbf{y}}(\mathbf{z} \mid e, \mathbf{y})$ is smooth with strictly positive density on its support.*

- *A2 (Conditional independence and content invariance): Given $(e, \mathbf{y})$, the coordinates of $\mathbf{z}$ are independent, $\log p_{\mathbf{z}|e,\mathbf{y}}(\mathbf{z} \mid e, \mathbf{y}) = \sum_{i=1}^{n_z} q_i(z_i; e, \mathbf{y})$,*

- *A3 (Linear independence): For any $\mathbf{b} \in \mathcal{B} \subseteq \mathbb{R}^{n_b}$ (and the relevant $\mathbf{y}$), there exist $n_b + 1$ environments $e_0, e_1, \ldots, e_{n_b}$ such that the $n_b$ vectors $\mathbf{w}(\mathbf{b}, e_j, \mathbf{y}) - \mathbf{w}(\mathbf{b}, e_0, \mathbf{y})$, $j = 1, \ldots, n_b$, are linearly independent, where $\mathbf{w}(\mathbf{b}, e_i, \mathbf{y}) := \left( \frac{\partial q_1(b_1; e_i, \mathbf{y})}{\partial b_1}, \ldots, \frac{\partial q_{n_b}(b_{n_b}; e_i, \mathbf{y})}{\partial b_{n_b}} \right)^\top.$*

- A4 (*Domain variability*): *There exist $e_i \neq e_j$ such that, for the relevant $\mathbf{y}$ and for any measurable set $A_{\mathbf{z}} \subseteq \mathcal{Z}$ with non-zero probability that cannot be written as $\Omega_{\mathbf{c}} \times \mathcal{B}$ for any $\Omega_{\mathbf{c}} \subset \mathcal{C}$, $\int_{A_{\mathbf{z}}} p(\mathbf{z} \mid \mathbf{e}_i, \mathbf{y}) \, d\mathbf{z} \neq \int_{A_{\mathbf{z}}} p(\mathbf{z} \mid \mathbf{e}_j, \mathbf{y}) \, d\mathbf{z}$.*

*Then the learned $\hat{\mathbf{c}}$ and $\hat{\mathbf{b}}$ are block-wise identifiable.*

**Definition 2.2.** (*Unblocked Influence*): Let $\mathbf{z}_1 \in \mathbb{R}^{d_1}$, $\mathbf{z}_2 \in \mathbb{R}^{d_2}$, and $\mathbf{z}_3 \in \mathbb{R}^{d_3}$ be random vectors. We say that unblocked influence for $\mathbf{z}_1$, $\mathbf{z}_2$ given $\mathbf{z}_3$ holds if there exists a causal path from $\mathbf{z}_1$ to $\mathbf{z}_3$ that is not blocked by $\mathbf{z}_2$. Formally, this condition requires $\mathbf{z}_1 \not\perp\!\!\!\perp \mathbf{z}_3 \mid \mathbf{z}_2$.

**Lemma 2.3.** *The bias $\mathbf{b}$ can be effectively utilized for prediction if Assumptions A1 through A4 hold and there is unblocked influence for $\mathbf{b}, y$ given $\mathbf{c}$.*

**When can bias be leveraged?** After identifying bias, we further provide theoretical insights into the conditions under which it can be leveraged to enhance predictive performance. Specifically, we show that the identified bias can contribute to prediction as long as there exists an active (i.e., unblocked) causal path from the bias to the prediction target, meaning the good information carried by the bias is not fully mediated or blocked by the content. As shown in Figure 2, there exists a path $y \to \mathbf{b}$ which indicates that $\mathbf{b}$ is not blocked with $y$ by $\mathbf{c}$. And it should be noted that even without this path, since $\mathbf{b} \leftarrow e \to y$, $\mathbf{b}$ is not blocked with $y$ by $\mathbf{c}$.

As shown in Lemma 2.1, under Assumptions A1–A4, both content and bias information $\mathbf{c}, \mathbf{b}$ can be well identified. It indicates that we can extract $\mathbf{c}, \mathbf{b}$ containing all the information as the ground-truth variables. Unblocked influence states that

$$P(y \mid \mathbf{c}, \mathbf{b}) \neq p(y \mid \mathbf{c}) \quad \text{on a set of nonzero measure.}$$

In Bayesian decision theory (for any proper loss, e.g. log-loss or squared error), the Bayes-optimal predictor using $(\mathbf{c}, \mathbf{b})$ is $f^*(\mathbf{c}, \mathbf{b}) = \arg\min_f \mathbb{E}\big[\ell\big(y, f(\mathbf{c}, \mathbf{b})\big)\big]$, which depends on the full conditional $p(y \mid \mathbf{c}, \mathbf{b})$. The best predictor using only $\mathbf{c}$ is $\arg\min_f \mathbb{E}\big[\ell(y, f(\mathbf{c}))\big]$, which depends only on $p(y \mid \mathbf{c})$. Since unblocked influence ensures that the conditional distributions $p(y \mid \mathbf{c})$ and $p(y \mid \mathbf{c}, \mathbf{b})$ differ, the joint predictor $f^*(\mathbf{c}, \mathbf{b})$ achieves a strictly lower expected loss compared to any predictor that relies solely on $\mathbf{c}$. This confirms that the bias variable $\mathbf{b}$ can indeed be exploited to improve predictive accuracy. A detailed proof, demonstrating that incorporating $\mathbf{b}$ leads to a better predictor under log-loss or squared error, is provided in Appendix B.3.

## 3 PRELIMINARY WORK ON BIAS CORRECTION

Recently, Stable Feature Boosting (SFB) (Eastwood et al., 2024) proposed a bias correction method that explores how bias can be leveraged for prediction. The core intuition is that predictions derived from invariant features are reliable across environments, and thus can serve as a reference signal for correcting the predictions from biased features. Concretely, SFB treats the predictions from invariant features as pseudo-labels and uses them to adjust the bias predictor, thereby reducing spurious correlations while still exploiting useful signals embedded in bias.

**Reliable invariant prediction assumption.** This assumption states that $p(y = 1 \mid \mathbf{c})$ remains stable across all domains, which indicates that $y \perp\!\!\!\perp e \mid \mathbf{c}$.

**Decomposition of Marginal Probability.** Specifically, according to Theorem 4.4 in (Eastwood et al., 2024), consider $\mathbf{b} \perp\!\!\!\perp \mathbf{c} \mid y$, the prediction process can be decomposed as follows:

$$p(y = 1 \mid \mathbf{c}, \mathbf{b}) = \sigma\bigg( \text{logit}\big(p(y = 1 \mid \mathbf{b})\big) + \text{logit}\big(p(y = 1 \mid \mathbf{c})\big) - \text{logit}\big(p(y = 1)\big) \bigg)$$

where $\sigma$ is the sigmoid function. By the reliable invariant prediction assumption, the predicted outputs using invariant features can serve as the pseudo labels in the correction process, and the label prior $p(y = 1)$ can be regarded as a constant in all domains.

**Correction with Pseudo Labels** Consider the predictor $f_c : \mathbf{c} \to y$. Define pseudo-label prediction as $\hat{y} = f_c(\mathbf{c})$. Let $h_0 = p(\hat{y} = 0 \mid y = 0), \quad h_1 = p(\hat{y} = 1 \mid y = 1)$. In the case that $\mathbf{c}$ contains

Figure 3: **Overall framework of the BAG method.** The framework consists of three main modules: representation learning, predictor training, and adaptation. In the representation learning stage, we employ a VAE to disentangle content and bias variables. For prediction, we construct a content predictor, a label prior, and a bias-aware predictor that reweights multiple domain experts using a domain estimator and also retrain under labels from content predictor in the test stage. These three predictor components work together for the final prediction.

the information for predicting $y$, it directly follows $h_0 + h_1 > 1$. Intuitively, if $c$ helps distinguish between $y = 0$ and $y = 1$, then a predictor based on $\mathbf{c}$ should be corrected more often than chance. Then, the correction of the bias predictor can be formulated as follows:

$$p(y = 1 \mid \mathbf{b}) = \frac{\Pr(\hat{y} = 1 \mid \mathbf{b}) + h_0 - 1}{h_0 + h_1 - 1}. \tag{1}$$

Since $\hat{y}$ is a function of $\mathbf{c}$, this correction relies on the assumption that $\mathbf{c}$ depends only on $y$, and the quantities $h_0$ and $h_1$ remain constant across all domains. Due to space constraints, we present only the core idea and method here, and refer readers to (Eastwood et al., 2024) and Appendix B for detailed derivations.

**Challenges.** The method relies on the assumption of reliable invariant prediction, i.e., $y \perp\!\!\!\perp e \mid \mathbf{c}$. However, as illustrated in Figure 2, this assumption no longer holds under label shift. In such cases, the prediction $p(y = 1 \mid \mathbf{b})$ is also affected by the confounding influence of $\mathbf{e}$. Ignoring $\mathbf{e}$ can therefore lead to suboptimal predictors.

## 4 METHOD

In this section, we introduce the Bias-Aware Generalization (**BAG**) framework, which systematically incorporates bias into the prediction process to improve OOD performance. The whole pipeline can be found in Figure 3. We start by learning disentangled representations so that $\mathbf{b}$ and $\mathbf{c}$ align with the data-generating process, producing factorized bias and content. Accordingly, we propose (i) Bias-Aware Prediction with an Environment Routing Mechanism and (ii) Bias Correction with an Adaptive Label Prior. For each component, we provide a theoretical analysis with a binary classification case and corresponding method details, while the multi-class extension and the full algorithm are deferred to Appendices C, D.1 and D.2.

### 4.1 REPRESENTATION LEARNING AND DISENTANGLEMENT

**Generative modeling.** Following Theorem 2.1, we take the observed data $\mathbf{x}$ from the source domains as input to the encoder, which maps it into the latent variable $\mathbf{z}$. We then decode $\mathbf{z}$ to reconstruct $\hat{\mathbf{x}}$ and train the encoder–decoder using the standard VAE loss:

$$\mathcal{L}_{\text{vae}} = \mathbb{E}_{\mathbf{x}}\big[\|\mathbf{x} - \hat{\mathbf{x}}\|^2 + \beta \text{KL}\big(q_\phi(\mathbf{z} \mid \mathbf{x}) \,\|\, p(\mathbf{z})\big)\big], \tag{2}$$

where $\beta$ is a hyper-parameter to balance the reconstruction loss and the KL divergence, which is set to 1 in our approach. The prior distribution $p(\mathbf{z})$ is chosen as an isotropic Gaussian. Minimizing Equation (2) encourages $(\mathbf{c}, \mathbf{b})$ to be identifiable in a block-wise sense and captures both invariant and domain-specific factors for further modeling.

**Regularization.** Following (Jiang & Veitch, 2022), we leverage a necessary but not sufficient condition for $\mathbf{c} \perp\!\!\!\perp \mathbf{b} \mid y$, that is $\mathbb{E}\big[\mathbf{c} \cdot (\mathbf{b} - \mathbb{E}[\mathbf{b} \mid y])\big] = 0$. Specifically, we add a regularization term to constrain the conditional independence, which can be written as:

$$\mathcal{L}_{ind} = \frac{1}{n} \sum_{i=1}^{n} \mathbf{c}_i \cdot \big(\mathbf{b}_i - \frac{1}{|\{j : y_j = y_i\}|} \sum_{j:y_j=y_i} \mathbf{b}_j\big).$$

After learning $\mathbf{b}$ and $\mathbf{c}$, our goal is to predict $y$ in the target domain via $p(y \mid \mathbf{b}, \mathbf{c})$.

### 4.2 BIAS-AWARE PREDICTION WITH ENVIRONMENT ROUTING MECHANISM

**Bias-aware predictors.** Only relying on content to guide the bias predictor has practical limitations: when $\mathbf{c}$ carries little information about $y$, the pseudo-labels $\hat{y}$ may be nearly random, making such guidance ineffective. Motivated by the causal path $y \leftarrow e \rightarrow \mathbf{b}$, we instead propose further leveraging bias to infer the environment $e$, and then using $e$ to route predictions. This improves $p(y \mid \mathbf{b})$ by explicitly modeling environmental context, independent of $\mathbf{c}$, and yields more robust bias utilization. The objective can be formulated as the following optimization task:

$$q^*(y \mid \mathbf{b}) = \arg\min_{q(\cdot \mid \mathbf{b})} \mathbb{E}_{e \sim p(e \mid \mathbf{b})}\big[D_{\mathrm{KL}}(p(y \mid \mathbf{b}, e) \,\|\, q(y \mid \mathbf{b}))\big]. \tag{3}$$

This suggests identifying a predictor that minimizes the expected Kullback-Leibler (KL) divergence from $p(y \mid \mathbf{b}, e)$ across all environments. The Bayes-optimal solution (Ohn & Lin, 2023) is

$$q^*(y \mid \mathbf{b}) = \sum_e p(e \mid \mathbf{b})\, p(y \mid \mathbf{b}, e)$$

This implies that we can leverage the identified bias to infer the environmental context and then develop a suitable bias-aware predictor $p(y \mid \mathbf{b})$ with a set of domain experts $p(y \mid \mathbf{b}, e)$ to bridge the gaps between environments. The detailed derivation can be found in Appendix B.4.

**Environment routing mechanism.** To exploit the bias $\mathbf{b}$ in an indirect manner, we introduce a set of $M$ learnable domain embeddings $\{e_1, e_2, \ldots, e_M\}$, which correspond to $M$ source domains $\{\mathcal{S}_1, \mathcal{S}_2, ..., \mathcal{S}_M\}$. Each embedding $e_k$ captures a distinct "mode" or style that may emerge in the source training data. Intuitively, this design follows the mixture-of-experts principle, allowing each expert to specialize in a subset of the training distribution.

In terms of model design, each $e_k$ is a learnable vector (or a small neural component) trained to represent a particular domain bias. We implement a domain classifier $p(e = e_i \mid \mathbf{b})$ via a softmax layer over linear outputs, to produce scalar weights.

These weights indicate how well each expert $e_i$ aligns with the domain-specific bias $\mathbf{b}$. For each expert, we parametrize $p(y \mid \mathbf{b}, e = e_i)$ as an MLP network, whose input is the combination of $\mathbf{b}$ and $e_i$. It models the expert specializing in domain $e_i$, making specific predictions based on the latent representation $\mathbf{b}$. Formally, we calculate the bias-aware predictor $f_b$ (i.e., $p(y \mid \mathbf{b})$) by reweighting domain-specific predictors:

$$f_b(\mathbf{b}) \;=\; \sum_{i=1}^{M} p\big(y \mid \mathbf{b}, e = e_i\big)\, p(e = e_i \mid \mathbf{b}). \tag{4}$$

### 4.3 BIAS CORRECTION WITH ADAPTIVE LABEL PRIOR

Existing bias correction methods, such as SFB (Eastwood et al., 2024), rely on content predictions as pseudo-labels to adjust the bias predictor under a fixed label prior. While effective in certain settings, this assumption breaks down under label shift, where the label distribution varies across environments. In such cases, a fixed prior leads to systematic miscalibration and limits the correction's effectiveness. To address this issue, we introduce an adaptive label prior that dynamically accounts for shifts in label distribution and identifies a new invariant component under label shift, thereby enabling more accurate and robust bias correction across domains. Since $p(y \mid \mathbf{c})$ is not invariant under label shift, we introduce an alternative decomposition to retain the invariant components while leveraging bias.

**Theorem 4.1** (Decomposition under label shift). *Given the data generation process illustrated in Figure 2, we have*

$$p(y = 1 \mid \mathbf{c}, \mathbf{b}) = \sigma \left( \underbrace{\mathrm{logit}\big(p(y = 1 \mid \mathbf{b})\big)}_{\textit{Bias Predictor}} + \underbrace{\log \frac{p(y = 1 \mid \mathbf{c})/p(y = 0)}{p(y = 0 \mid \mathbf{c})/p(y = 1)}}_{\textit{Invariant Predictor}} \right) \quad (5)$$

*Due to the label shift, the content prediction $p(y \mid \mathbf{c})$ is not invariant, and the label prior $p(y)$ is not a constant. Interestingly, we found that the composition of both of them is invariant.*

**Discussion on invariant prediction.** Although $p(y \mid \mathbf{c})$ is not invariant, we observe that $p(\mathbf{c} \mid y)$ remains invariant across domains, since $y$ d-separate $\mathbf{c}$ and the environment variable $e$. Formally, $\forall e_i, e_j \in E_{all}$, since $\mathbf{c}$ only depends on $y$, it follows that

$$p(\mathbf{c} \mid y = 1, e = e_i) = p(\mathbf{c} \mid y = 1, e = e_j) \Rightarrow \frac{p(y = 1, e = e_i \mid \mathbf{c})}{p(y = 1, e = e_i)} = \frac{p(y = 1, e = e_j \mid \mathbf{c})}{p(y = 1, e = e_j)}$$

which implies $\frac{p(y|\mathbf{c})}{p(y)}$ is an invariant quantity across all domains. It motivates us to jointly learn the content predictor and the learnable label prior. The detailed proof can be found in Appendix B.5.

**Predictors** Guided by Theorem 4.1, we implement the label predictor into two components:

$$f(\mathbf{z}) = f_b(\mathbf{b}) + f_{inv}(\mathbf{c}, Pr),$$

where $f_b$ denotes the bias-aware predictor $p(y \mid \mathbf{b})$, and $f_{inv}$ is the invariant predictor built on the content variable $\mathbf{c}$ and the learnable prior term $Pr = \mathrm{logit}\big(p(y)\big)$. In detail, $f_{in}(\mathbf{c}, Pr) = f_c(\mathbf{c}) - Pr$ where $f_c(\mathbf{c}) = p(y \mid \mathbf{c})$. In particular, $f_{inv}$ is an "invariant" module across all domains and can be used directly in the target domain without further modification, even under label shift. In training, we jointly learn bias predictors, the content predictor, and the logits prior with a cross-entropy classification loss $\mathcal{L}_{\mathrm{cls}}$.

**Theorem 4.2** (Upper bound of Bias Correction). *Under the data generation process in Figure 2, define the corrected bias predictor $\widehat{f}(\mathbf{b}) = \frac{\widehat{q}(\mathbf{b}) + \hat{h}_0 - 1}{\hat{h}_0 + \hat{h}_1 - 1}$, where $\widehat{q}(\mathbf{b})$ is the estimator of $q(\mathbf{b}) = p(\hat{y} = 1 \mid \mathbf{b})$, $\hat{h}_0, \hat{h}_1$ are estimators of $h_0$ and $h_1$. Let $f^\star(\mathbf{b}) = p(y = 1 \mid \mathbf{b})$ mean using only the $\mathbf{b}$ learned on sources to predict $y$ at the test stage. For the target cross-entropy risk on the bias subproblem, define: $\mathcal{R}_{(b)}(g) = \mathbb{E}_{\mathbf{b} \sim p(\mathbf{b})}\Big(\mathrm{KL}\big(\mathrm{Bern}(p(y = 1 \mid \mathbf{b})) \,\|\, \mathrm{Bern}(g(\mathbf{b}))\big)\Big)$. Then there exist constants $c_1, c_2 > 0$ (depending only on the margin assumption) such that:*

$$\mathcal{R}_{(b)}(\widehat{f}) \leq \frac{c_1}{(h_0 + h_1 - 1)^2} \mathbb{E}\big[(\widehat{q}(\mathbf{b}) - q(\mathbf{b}))^2\big] + \frac{c_2}{(h_0 + h_1 - 1)^2} \left((\hat{h}_0 - h_0)^2 + (\hat{h}_1 - h_1)^2\right) \quad (6)$$

*In particular, if this upper bound is smaller than $\mathcal{R}_{(b)}(f^\star)$ (equivalently, $\Delta_{\mathrm{DG}}$), then $\widehat{f}$ attains strictly lower risk on the target domain than $f^\star$.*

Theorem 4.2 shows that while correction incurs small statistical costs from estimating $q(\mathbf{b})$ and $(h_0, h_1)$, these diminish with data. So under the data generation process in Figure 2, the corrected predictor is provably more reliable than source-only deployment (see Appendix B.7 for details).

**Bias adaptation.** In the training phase, the content predictor $f_c$ and the learnable label prior $Pr$ are jointly optimized to form the invariant component, which is then frozen for the test phase. During test time, given test-domain inputs $\{\mathbf{x}^\mathcal{T}\}$, we first extract the latent representations $[\mathbf{c}^\mathcal{T}, \mathbf{b}^\mathcal{T}]$ using the pre-trained encoder. We then leverage the learned content predictor $f_c(\mathbf{c}^\mathcal{T})$ to generate pseudo-labels $\hat{y}^\mathcal{T}$ on unlabeled test data. In the source domains, we collect $h_0$ and $h_1$, which are entries of the confusion-matrix probabilities between pseudo-labels and ground truth. Intuitively, these parameters adjust how domain-invariant information maps the bias $\mathbf{b}$ to valid probability estimates in the new domain. To estimate $p(\hat{y} = 1 \mid \mathbf{b})$, we update the bias-aware predictor $f_b$ with the generated pseudo-labels $\hat{y}^\mathcal{T}$ as

$$\min_{f_b} \mathbb{E}_{\mathbf{b}^\mathcal{T}} \Big[\mathcal{L}_{ada} = \ell\Big(\sigma\big(f_b(\mathbf{b}^\mathcal{T})\big), \hat{y}\Big)\Big], \quad (7)$$

where the adaptation loss $\mathcal{L}_{ada}$ is characterized as the cross-entropy loss with pseudo labels. We record this post-trained predictor as $\tilde{f}_b(\mathbf{b}^{\mathcal{T}})$. Let $\phi(\cdot) = \text{logit}((\frac{\sigma(\cdot)+h_0-1}{h_0+h_1-1}))$ represent the correction function. The final prediction is

$$f([\mathbf{c}^{\mathcal{T}}, \mathbf{b}^{\mathcal{T}}]) = f_c(\mathbf{c}^{\mathcal{T}}) + \phi(\tilde{f}_b(\mathbf{b}^{\mathcal{T}})) - Pr.$$

**Objectives.**    Overall, the losses in both the training and bias adaptation stages are as follows:

$$\text{Training Stage:} \quad \mathcal{L}_{all} = \mathcal{L}_{cls} + \lambda_0 \mathcal{L}_{\text{vae}} + \lambda_1 \mathcal{L}_{ind}$$
$$\text{Adaptation Stage:} \quad \mathcal{L}_{ada}$$

where $\lambda_0, \lambda_1$ are the hyperparameters.

## 5 EXPERIMENTS

In this section, we test **BAG** on both synthetic data and real-world datasets, following the setting in ACTIR (Jiang & Veitch, 2022) and SFB (Eastwood et al., 2024), for a fair comparison.

**Baseline methods.**    We compare our method with ERM and IRM (Arjovsky et al., 2019), a classic approach that leverages invariant information for making predictions. Additionally, we compare with ACTIR (Jiang & Veitch, 2022), which disentangles the invariant feature and bias by independent regularization. And with SFB (Eastwood et al., 2024), which aims to exploit spurious features but in a weaker setting. SFB can serve as a special case of our method when 1) the environment doesn't affect the label, and 2) it's not bias-aware for prediction. For the Office-Home dataset, since SFB doesn't have results on it, we add MMD (Li et al., 2018a) and GMDG (Tan et al., 2024) as more baselines to show our performance. For the DomianNet, we also add DANN (Ganin et al., 2016b), CDANN (Long et al., 2018), SagNet (Nam et al., 2021), EQRM (Eastwood et al., 2022), RDM (Nguyen et al., 2024) as more baselines.

### 5.1 SIMULATION EXPERIMENTS

**Data simulation.**    To examine whether **BAG** can effectively identify and leverage domain shift information for better generalization, we devised a series of simulation experiments. Our data generation process strictly adheres to the causal structure depicted in Figure 2. First, we sample an environmental variable $e$ following a categorical distribution, $e \sim \text{Categorical}(\{\pi_1, \ldots, \pi_M\})$, which subsequently induces variations across different domains. Then, a binary label $y$ is generated in an environment-dependent manner: $y = \mathbf{1}\{ w^{\mathsf{T}}E + b_0 > 0 \}$, where $E$ denotes an embedding of $e$.

Next, we assign stable content $c$ based on the value of $y$: if $y = 0$, then $c = c_0 + \epsilon$; if $y = 1$, then $c = c_1 + \epsilon$. Here, $\epsilon \sim \mathcal{N}(0, \sigma_c^2 I)$. Simultaneously, the bias variable $b$ is determined as: $b = E + \mathbf{C}_{e,y}$, ensuring that the background shift is influenced jointly by the environment and the label. This setup emulates the common real-world scenario where false associations often arise. Finally, stable content and bias jointly generate the observed data: $x = M \begin{bmatrix} c \\ b \end{bmatrix} + \eta$, where $M$ is an invertible matrix, and $\eta \sim \mathcal{N}(0, \sigma_x^2 I)$ represents measurement noise.

**Results and Discussions.**    This study evaluates several methods on a synthetic dataset of 5000 samples. To ensure result reliability, each method was repeated five times, and the average performance was recorded. As shown on the right side of Table 2, both ERM and IRM performed well on the training set but achieved test accuracies just above 50%. We attribute this to the significant distributional disparity of environmental information $e$ across domains in the dataset, which limits the effectiveness of the class label in supporting accurate predictions. In other words, an excessive focus on invariant information may hinder model optimization, leading to marginal performance gains. In contrast, our method leverages domain-specific information ($\mathbf{b}$) more effectively, significantly enhancing prediction performance by reducing noise relative to relying solely on the class label.

### 5.2 REAL-WORLD EXPERIMENTS

**Experimental Settings.**    We test our approach in commonly used PACS (Li et al., 2017) and Office-Home (Venkateswara et al., 2017) datasets. In these two datasets, we let one domain become

Table 2: Comparison on the PACS dataset and the Synthetic dataset. All **BAG** results on PACS are obtained by averaging over 3 seeds. Baseline results are taken from (Eastwood et al., 2024).

| Algorithm | PACS | | | | | Synthetic |
|---|---|---|---|---|---|---|
| | P | A | C | S | Avg | Acc |
| ERM | 93.0±0.7 | 79.3±0.5 | 74.3±0.7 | 65.4±1.5 | 78.0 | 54.62±3.70 |
| ERM + PL | 93.7±0.4 | 79.6±1.5 | 74.1±1.2 | 63.1±3.1 | 77.6 | – |
| IRM (Arjovsky et al., 2019) | 93.3±0.3 | 78.7±0.7 | 75.4±1.5 | 65.6±2.5 | 78.3 | 54.73±2.72 |
| IRM + PL | 94.1±0.7 | 78.9±2.9 | 75.1±4.6 | 62.9±4.9 | 77.8 | – |
| ACTIR (Jiang & Veitch, 2022) | 94.8±0.1 | 82.5±0.4 | 76.6±0.6 | 62.1±1.3 | 79.0 | 70.84±6.37 |
| SFB (Eastwood et al., 2024) | 95.8±0.6 | 80.4±1.3 | 76.6±0.6 | 71.8±2.0 | 81.2 | 82.65±3.53 |
| GMDG (Tan et al., 2024) | 96.6±0.7 | 83.9±0.4 | 75.4±1.7 | 70.1±3.2 | 81.5 | – |
| **BAG** | **96.6**±0.8 | **86.0**±0.4 | **77.9**±1.0 | **73.0**±0.4 | **83.4** | **97.48**±1.82 |

Table 3: DomainNet target-domain accuracies (%). All methods use ResNet-50 as backbones. Most baseline results are taken from Eastwood et al. (2022).

| Algorithm | clipart | infograph | painting | quickdraw | real | sketch | Avg |
|---|---|---|---|---|---|---|---|
| MIRO (Cha et al., 2022) | – | – | – | – | – | – | 44.3 |
| RDM (Nguyen et al., 2024) | – | – | – | – | – | – | 43.4 |
| ERM | 58.1 | 18.8 | 46.7 | 12.2 | 59.6 | 49.8 | 40.9 |
| DANN (Ganin et al., 2016b) | 53.1 | 18.3 | 44.2 | 11.8 | 55.5 | 46.8 | 38.3 |
| MMD (Li et al., 2018a) | 32.1 | 11.0 | 26.8 | 8.7 | 32.7 | 28.9 | 23.4 |
| CDANN (Long et al., 2018) | 54.6 | 17.3 | 43.7 | 12.1 | 56.2 | 45.9 | 38.3 |
| IRM(Arjovsky et al., 2019) | 48.5 | 15.0 | 38.3 | 10.9 | 48.2 | 42.3 | 33.9 |
| SagNet (Nam et al., 2021) | 57.7 | 19.0 | 45.3 | 12.7 | 58.1 | 48.8 | 40.3 |
| EQRM (Eastwood et al., 2022) | 56.1 | 19.6 | 46.3 | 12.9 | 61.1 | 50.3 | 41.0 |
| ACTIR (Jiang & Veitch, 2022) | 50.0 | 22.6 | 43.5 | 11.7 | 57.8 | 46.8 | 38.7 |
| GMDG (Tan et al., 2024) | **63.4** | 22.4 | 51.4 | 13.4 | 64.4 | **52.4** | 44.6 |
| BAG | 61.8 | **25.6** | **51.8** | **13.9** | **65.5** | 50.3 | **44.8** |

the target and the other domains as source domains. We use ResNet-18 for PACS and ResNet-50 for Office-Home as backbones, with MLP-based VAEs and classifiers. We report results averaged over 3 random seeds. Our framework employs a single-layer and a two-layer linear layer as the encoder and decoder, respectively, and utilizes a single-layer linear layer for each specific classifier. We define three learnable parameters for the environment embedding . The details can be found in Appendix E.1. For DomainNet, we also use Resnet-50 as backbones.

**Results and Discussions.** For the PACS data, on the left side of Table 2, we can find that our model is far superior to other models, with an average improvement of 2.2% compared with the best baseline, SFB method. Our model shows consistent improvement across all target domains. Among all target domains, we achieved better improvement on the Art Painting test, which may be because of the related bias in the domains. According to the experimental results of the Office-

Table 1: Results on Office-Home with ResNet-50. BAG results are averaged over 3 seeds.

| Algorithm | Office-Home | | | | |
|---|---|---|---|---|---|
| | Art | Clipart | Product | Real | Avg |
| ERM | 63.1 | 51.9 | 77.2 | 78.1 | 67.6 |
| IRM (Arjovsky et al., 2019) | 62.4 | 53.4 | 75.5 | 77.7 | 67.2 |
| SFB (Eastwood et al., 2024) | – | – | – | – | – |
| MMD (Li et al., 2018a) | 62.4 | 53.6 | 75.8 | 76.4 | 67.1 |
| ACTIR (Jiang & Veitch, 2022) | 67.2 | 55.5 | 78.7 | 81.1 | 70.6 |
| GMDG (Tan et al., 2024) | **68.9** | 56.2 | 79.9 | 82.0 | 71.7 |
| **BAG** | 68.8 | **57.2** | **80.0** | **82.1** | **72.0** |

Home dataset on the Table 1, our **BAG** model outperforms all other baselines on three transfer tasks. It should be noted that we achieve much better performance in Clipart, which is the harder task; it shows **BAG** can achieve better generalization even when the domain gap is larger than the baselines. Our method attains the best average accuracy on DomainNet (44.8%), surpassing GMDG (44.6%) and MIRO (44.3%). It achieves the per-domain best on infograph, painting ,quickdraw, andreal , while remaining competitive on clipart and sketch.The largest gain occurs on infograph, where background/stylistic artifacts are prominent, suggesting that disentangling content from bias

effectively mitigates spurious-domain cues and furthur help to use bias. On clipart and sketch we are not SOTA, which is likely because the domain-specific bias signal in these two domains is weak, uninformative, or misaligned, limiting the benefit of leveraging it. Compared with ACTIR under the same setting, our method achieves substantially higher averages, which we attribute to more accurate disentanglement of, and principled use of, bias. Considering that **BAG** involves two steps of optimization and one VAE , we demonstrate the effects of each individual optimization strategy and usage of VAE by ablation study. Please note that "-" in followings names means that, in the ablation study, we drop and only drop the corresponding part. We have **BAG-VAE** means drop the **VAE** loss and just use a linear layer, **BAG-TTA** means we drop the **T**est **T**ime **A**daptation of using persudo label to guide bias part, **BAG-RE** means that we drop the **R**eweight **E**xperts part of our model. We show that all the models perform better compared with baselines both in PACS and Synthetic data, but worse than **BAG**. Details of the ablation study can be found in Appendix E.2.

## 6  CONCLUSION

In this paper, we challenge the conventional methods for OOD generalization that treat bias solely as an obstacle. Instead, we theoretically demonstrate that bias can be strategically leveraged to enhance prediction, particularly when it retains useful dependencies on the target domain labels. Our framework demonstrates that the bias can be utilized in two ways. Through the empirical validation on synthetic and real-world benchmarks, we establish the effectiveness of this approach. The results consistently show that leveraging bias in a structured manner leads to more robust and adaptable models, offering a new perspective on OOD generalization. **Limitations.** This work primarily aims to question the conventional treatment of bias and provide a theoretical foundation for understanding its effects. The experiments presented in this paper are intended to validate our proposed method and do not include experiments on large-scale datasets. We acknowledge this limitation and leave systematic benchmarking on large-scale datasets as an important direction for future work.

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

# A RELATED WORK

**Domain generalization** Domain Generalization (DG) aims to develop models that generalize to unseen OOD target domains using only observed source domains for training (Zou et al., 2024; Zhang et al., 2024; Chen et al., 2023). A common approach is learning invariant representations across all source domains, as seen in Invariant Risk Minimization (IRM) (Arjovsky et al., 2019) and its extensions (Ahuja et al., 2020; Rosenfeld et al., 2020; Ahuja et al., 2021; Kamath et al., 2021; Krueger et al., 2021). Other methods leverage adversarial learning (Li et al., 2018b; Shao et al., 2019; Zhao et al., 2020; Deng et al., 2020; Matsuura & Harada, 2020) to extract domain-invariant features or use disentangled representation learning (Ilse et al., 2020; Wang et al., 2020b; Jiang & Veitch, 2022) to separate domain-invariant from domain-specific information. Causal inference techniques (Wang et al., 2020c; Qi et al., 2020), such as back-door and front-door rules, help mitigate bias. Furthermore, techniques such as data augmentation (Volpi & Murino, 2019; Shi et al., 2020; Huang et al., 2021; Li et al., 2021), feature augmentation (Mancini et al., 2020; Shu et al., 2021; Zhou et al., 2024), and network regularization (Seo et al., 2020; Segu et al., 2023) can reduce overfitting and improve generalization.

**Causal Representation Learning** Causal Representation Learning (CRL) aims to recover latent variables that correspond to the causal generative factors of high-dimensional observations and the mechanisms that link them (Schölkopf et al., 2021). Building on the invariance principle, Invariant Causal Prediction formalizes causal variable selection via stability across environments (Peters et al., 2016). A rich line of identifiability results shows that with auxiliary variables or weak interventions, nonlinear ICA and VAE-based approaches can recover latent sources up to permissible transformations (Hyvarinen et al., 2019; Khemakhem et al., 2020). For vision, structured VAEs with causal priors explicitly separate content from style/context (Yang et al., 2021), and temporal–interventional setups such as Citris leverage intervention sequences to learn identifiable factors (Lippe et al., 2022). Beyond model proposals, empirical and theoretical analyses caution that purely unsupervised disentanglement is generally impossible without inductive biases or weak supervision, motivating the use of environment cues or metadata (Locatello et al., 2019). One line of research leverages distributional shifts in latent variables, where nonstationary or environmental variation facilitates the recovery of nonlinear independent components and the underlying causal structures (Kong et al., 2022a; Zhang & Hyvärinen, 2011; Chen et al., 2024). Our approach follows this CRL perspective: we infer a latent space with a VAE, factorize it into invariant content and bias/context, then exploit the bias variables for environment-aware prediction while aligning them with invariant predictions.

**Label shift in OOD problems** Label shift occurs when the label prior $p(y)$ changes between training and test domains while the class-conditional $p(x \mid y)$ remains stable (Lipton et al., 2018). Classical corrections include EM-based prior adjustment (Saerens et al., 2002) and Black-Box Shift Estimation (BBSE), which uses a calibrated source classifier's confusion matrix to estimate target priors and reweight predictions (Lipton et al., 2018). Subsequent work unifies moment-matching and maximum-likelihood estimators and emphasizes the central role of calibration (e.g., bias-corrected temperature scaling, BCTS) for reliable prior estimation (Garg et al., 2020; Azizzadenesheli et al., 2019). In streaming or nonstationary settings, online label-shift adaptation updates class-prior estimates on the fly without access to target labels (Wu et al., 2021; Bai et al., 2022). Benchmarks such as WILDS and Wild-Time document that label shift frequently co-occurs with input shift in real applications (e.g., medical imaging, wildlife monitoring, and temporally evolving text) (Koh et al., 2021; Yao et al., 2022). Consistent with this evidence, we explicitly account for changes in $p(y)$ when adapting to new environments, combining prior correction with environment-aware mixture-of-experts weighting informed by the biased latent variables.

**Unsupervised domain adaptation** Unlike DG, unsupervised domain adaptation utilizes unlabeled target data to guide model adaptation. Due to the observation of target distribution, many methods focus on distribution alignment by minimizing domain divergence (Rozantsev et al., 2018; Sun & Saenko, 2016; Kang et al., 2019; Liu et al., 2020), using importance reweighting (Jiang & Zhai, 2007; Xu et al., 2019; Fang et al., 2020), and learning with adversarial discriminators, such as DANN (Ganin et al., 2016a), ADDA (Tzeng et al., 2017), and WDDA (Shen et al., 2018). Beyond distribution alignment, some methods focus on invariant information-based adaptation, including self-training (Lee et al., 2013; Zou et al., 2018; Saito et al., 2020; Chen et al., 2020; Kumar et al., 2020; Liu et al., 2021a), which uses pseudo-labels from a source-trained model to guide target domain learning, and batch normalization-based regularization (Li et al., 2018c; Liu et al., 2021b), which

adapts only BN layers while keeping other parameters invariant. Recently, disentangled representation learning (Kong et al., 2022b; Li et al., 2024b) has shown strong potential for adaptation by separating domain-invariant content from domain-specific bias.

**Test-Time domain adaptation** In real-world scenarios, the target domain distribution is often unknown in advance, making it challenging to train an adaptive model beforehand, which motivates test time adaptation. The typical solution is Test-Time Training (TTT) (Sun et al., 2020; Liu et al., 2021c; Mirza et al., 2023; Li et al., 2023a), which adapts models during inference by optimizing the same self-supervised learning objective used during pre-training. Besides, even without access to the target distribution, adaptation remains feasible with invariant information learned from the source domain, such as through self-training (Chen et al., 2022; Tomar et al., 2023; Su et al., 2022; Goyal et al., 2022) or network regularization (Wang et al., 2020a; Liang et al., 2020). The most closely related work to ours is Stable Feature Boosting (SFB), which leverages unstable features alongside invariant representations to generate pseudo-labels when only the image is affected by environmental factors. Our approach differs in three key ways. First, we delve deeper into latent variable modeling to provide a more comprehensive explanation of bias. Second, our model accommodates more complex bias scenarios, allowing the environment variable to influence both the image and the label. Third, we integrate invariant correction and expert reweighting to effectively utilize biased data.

# B PROOF AND DISCUSSION OF SECTION 2

## B.1 PROOF OF THE IDENTIFICATION OF LATENT VARIABLES

**Lemma B.1.** *(**Block-wise identification of c and b**; cf. Kong et al. (2023).) Assume the data-generating process in Figure 2: the observation is $\mathbf{x} = g(\mathbf{z})$ with $\mathbf{z} = (\mathbf{b}^\top, \mathbf{c}^\top)^\top$, where $g$ is an invertible diffeomorphism. Let $\hat{\mathbf{x}} = \hat{g}(\hat{\mathbf{z}})$ be another invertible generator with latent $\hat{\mathbf{z}} = (\hat{\mathbf{b}}^\top, \hat{\mathbf{c}}^\top)^\top$, and suppose the model matches the conditional distribution $p_{\hat{\mathbf{x}}|\mathbf{e},\mathbf{y}} = p_{\mathbf{x}|\mathbf{e},\mathbf{y}}$ for all $(\mathbf{e}, \mathbf{y})$. Assume:*

- *A1 (Smooth and positive density): $p_{\mathbf{z}|\mathbf{e},\mathbf{y}}(\mathbf{z} \mid \mathbf{e}, \mathbf{y})$ is smooth with strictly positive density on its support.*

- *A2 (Conditional independence and content invariance): Given $(\mathbf{e}, \mathbf{y})$, the coordinates of $\mathbf{z}$ are independent,*

$$\log p_{\mathbf{z}|\mathbf{e},\mathbf{y}}(\mathbf{z} \mid \mathbf{e}, \mathbf{y}) = \sum_{i=1}^{n} q_i(z_i; \mathbf{e}, \mathbf{y}),$$

*where $n = n_b + n_c$, $z_1, \ldots, z_{n_b}$ correspond to $\mathbf{b}$ and $z_{n_b+1}, \ldots, z_n$ correspond to $\mathbf{c}$. Moreover, content coordinates are environment-invariant: for $i > n_b$, $q_i(c_{i-n_b}; \mathbf{e}, \mathbf{y}) = q_i(c_{i-n_b}; \mathbf{y})$.*

- *A3 (Linear independence): For any $\mathbf{b} \in \mathcal{B} \subseteq \mathbb{R}^{n_b}$ (and the relevant $\mathbf{y}$), there exist $n_b + 1$ environments $e_0, e_1, \ldots, e_{n_b}$ such that the $n_b$ vectors*

$$\boldsymbol{w}(\mathbf{b}, e_j, \mathbf{y}) - \boldsymbol{w}(\mathbf{b}, e_0, \mathbf{y}), \quad j = 1, \ldots, n_b,$$

*are linearly independent, where*

$$\boldsymbol{w}(\mathbf{b}, e_i, \mathbf{y}) := \left( \frac{\partial q_1(b_1; e_i, \mathbf{y})}{\partial b_1}, \ldots, \frac{\partial q_{n_b}(b_{n_b}; e_i, \mathbf{y})}{\partial b_{n_b}} \right)^\top.$$

- *A4 (Domain variability): There exist $e_i \neq e_j$ such that, for the relevant $\mathbf{y}$ and for any measurable set $A_{\mathbf{z}} \subseteq \mathcal{Z}$ with non-zero probability that cannot be written as $\Omega_{\mathbf{c}} \times \mathcal{B}$ for any $\Omega_{\mathbf{c}} \subset \mathcal{C}$,*

$$\int_{A_{\mathbf{z}}} p(\mathbf{z} \mid e_i, \mathbf{y}) \, d\mathbf{z} \neq \int_{A_{\mathbf{z}}} p(\mathbf{z} \mid e_j, \mathbf{y}) \, d\mathbf{z}.$$

*Define the learned invariant block $\hat{\mathbf{c}}$ to be those coordinates of $\hat{\mathbf{z}}$ whose conditional distribution does not vary with the environment: $p_{\hat{c}_j|\mathbf{e},\mathbf{y}}$ is independent of $\mathbf{e}$ for all $j$ (and let the remaining coordinates form $\hat{\mathbf{b}}$). Then the learned $(\hat{\mathbf{c}}, \hat{\mathbf{b}})$ are block-wise identifiable, i.e., there exist invertible functions $h_c$ and $h_b$ (up to permutations and coordinate-wise reparameterizations) such that $\mathbf{c} = h_c(\hat{\mathbf{c}})$ and $\mathbf{b} = h_b(\hat{\mathbf{b}})$.*

*Proof.* We start from the matched conditional distribution $p_{\hat{\mathbf{x}}|\mathbf{e},\mathbf{y}} = p_{\mathbf{x}|\mathbf{e},\mathbf{y}}$ and introduce the diffeomorphism

$$h := g^{-1} \circ \hat{g} : \ \hat{\mathbf{z}} \mapsto \mathbf{z}.$$

Then, for all $(\mathbf{e}, \mathbf{y})$,

$$p_{\hat{g}(\hat{\mathbf{z}})|\mathbf{e},\mathbf{y}} = p_{g(\mathbf{z})|\mathbf{e},\mathbf{y}}. \tag{8}$$

By the change-of-variables formula,

$$p_{\hat{\mathbf{z}}|\mathbf{e},\mathbf{y}}(\hat{\mathbf{z}}) = p_{\mathbf{z}|\mathbf{e},\mathbf{y}}\big(h(\hat{\mathbf{z}})\big) \, \big| \det \mathbf{J}_h(\hat{\mathbf{z}}) \big|, \tag{9}$$

where $\mathbf{J}_h(\hat{\mathbf{z}})$ is the Jacobian of $h$ at $\hat{\mathbf{z}}$. Since $g, \hat{g}$ are diffeomorphisms, $h$ is invertible and $\det \mathbf{J}_h(\hat{\mathbf{z}}) \neq 0$.

By A2 (conditional independence), letting $q_i(z_i; \mathbf{e}, \mathbf{y}) := \log p_{z_i|\mathbf{e},\mathbf{y}}(z_i)$ and $\hat{q}_i(\hat{z}_i; \mathbf{e}, \mathbf{y}) := \log p_{\hat{z}_i|\mathbf{e},\mathbf{y}}(\hat{z}_i)$, we have

$$\log p_{\mathbf{z}|\mathbf{e},\mathbf{y}}(\mathbf{z} \mid \mathbf{e}, \mathbf{y}) = \sum_{i=1}^{n} q_i(z_i; \mathbf{e}, \mathbf{y}), \quad \log p_{\hat{\mathbf{z}}|\mathbf{e},\mathbf{y}}(\hat{\mathbf{z}} \mid \mathbf{e}, \mathbf{y}) = \sum_{i=1}^{n} \hat{q}_i(\hat{z}_i; \mathbf{e}, \mathbf{y}). \tag{10}$$

Combining equation 9–equation 10 and writing $\mathbf{z} = h(\hat{\mathbf{z}})$ gives

$$\sum_{i=1}^{n} q_i(z_i; \mathbf{e}, \mathbf{y}) + \log \big| \det \mathbf{J}_h(\hat{\mathbf{z}}) \big| = \sum_{i=1}^{n} \hat{q}_i(\hat{z}_i; \mathbf{e}, \mathbf{y}). \tag{11}$$

**Step 1: $\mathbf{B} = \partial \mathbf{b}/\partial \hat{\mathbf{c}} = \mathbf{0}$.** Differentiate equation 11 with respect to $\hat{c}_j$ for any $j \in \{1, \ldots, n_c\}$:

$$\sum_{i=1}^{n} \frac{\partial q_i(z_i; \mathbf{e}, \mathbf{y})}{\partial z_i} \cdot \frac{\partial z_i}{\partial \hat{c}_j} + \frac{\partial}{\partial \hat{c}_j} \log \big| \det \mathbf{J}_h(\hat{\mathbf{z}}) \big| = \frac{\partial \hat{q}_j(\hat{c}_j; \mathbf{e}, \mathbf{y})}{\partial \hat{c}_j}. \tag{12}$$

Note that $\mathbf{J}_h$ does not depend on $\mathbf{e}$, since $h = g^{-1} \circ \hat{g}$ is a mapping between latent spaces only. Fix $e_0, \ldots, e_{n_b}$ as in A3 and subtract the version of equation 12 at $e_0$ from that at $e_k$ ($k = 1, \ldots, n_b$); the Jacobian term cancels:

$$\sum_{i=1}^{n} \left( \frac{\partial q_i(z_i; e_k, \mathbf{y})}{\partial z_i} - \frac{\partial q_i(z_i; e_0, \mathbf{y})}{\partial z_i} \right) \cdot \frac{\partial z_i}{\partial \hat{c}_j} = \frac{\partial \hat{q}_j(\hat{c}_j; e_k, \mathbf{y})}{\partial \hat{c}_j} - \frac{\partial \hat{q}_j(\hat{c}_j; e_0, \mathbf{y})}{\partial \hat{c}_j}. \tag{13}$$

By A2, for content coordinates ($i > n_b$) the difference on the left is zero because $q_i$ does not depend on $\mathbf{e}$. By the definition of the learned invariant block, $p_{\hat{c}_j|\mathbf{e},\mathbf{y}}$ is invariant in $\mathbf{e}$, hence the right-hand side of equation 13 is zero. Therefore,

$$\sum_{i=1}^{n_b} \left( \frac{\partial q_i(b_i; e_k, \mathbf{y})}{\partial b_i} - \frac{\partial q_i(b_i; e_0, \mathbf{y})}{\partial b_i} \right) \cdot \frac{\partial b_i}{\partial \hat{c}_j} = 0, \quad k = 1, \ldots, n_b.$$

By A3, the $n_b$ vectors $\boldsymbol{w}(\mathbf{b}, e_k, \mathbf{y}) - \boldsymbol{w}(\mathbf{b}, e_0, \mathbf{y})$ are linearly independent, so the only solution is

$$\frac{\partial b_i}{\partial \hat{c}_j} = 0, \qquad \forall i \in \{1, \ldots, n_b\}, \ j \in \{1, \ldots, n_c\}.$$

Let the Jacobian of $h$ be block-partitioned as

$$\mathbf{J}_h = \begin{bmatrix} \mathbf{A} & \mathbf{B} \\ \mathbf{C} & \mathbf{D} \end{bmatrix} = \begin{bmatrix} \partial \mathbf{b}/\partial \hat{\mathbf{b}} & \partial \mathbf{b}/\partial \hat{\mathbf{c}} \\ \partial \mathbf{c}/\partial \hat{\mathbf{b}} & \partial \mathbf{c}/\partial \hat{\mathbf{c}} \end{bmatrix}.$$

The conclusion above yields $\mathbf{B} = \mathbf{0}$. Since $h$ is invertible, $\mathbf{J}_h$ has full rank, and therefore there exists a function $h_b$ such that $\mathbf{b} = h_b(\hat{\mathbf{b}})$.

**Step 2: $\mathbf{C} = \partial \mathbf{c}/\partial \hat{\mathbf{b}} = \mathbf{0}$.** If $\mathbf{C} \neq \mathbf{0}$, then changes in $\hat{\mathbf{b}}$ would induce changes in $\mathbf{c}$. Combining A4 (which ensures that environment variation manifests on non-product measurable sets) with positivity and invertibility would lead to a contradiction with the matched conditional distribution in equation 8. A detailed argument follows (Kong et al., 2023, Theorem 4.2, Steps 1–3). Hence $\mathbf{C} = \mathbf{0}$, and by full rank there exists $h_c$ such that $\mathbf{c} = h_c(\hat{\mathbf{c}})$.

Therefore, $\mathbf{J}_h$ is block-diagonal, which establishes block-wise identifiability of $(\hat{\mathbf{c}}, \hat{\mathbf{b}})$. $\qquad \square$

## B.2 Discussion of Lemma 2.1

**Concepts Explanation** We would like to detailedly explain some concepts in Lemma 2.1 as follows: (1)Here "Positive support" means the set of all points where a probability distribution assigns strictly positive probability or density (i.e., $p(x) > 0$). This is emphasized because the probability distribution may have some point in 0 for example the distribution are uniform distribution. To maintain academic rigor, this hypothesis was highlighted. (2)The $\mathcal{B}$ is introduced in A3 and it refers to the set (or space) of domain-specific latent variables. (3)Linear independence means that changes in different environments produce distinct, non-redundant effects on the domain-specific latent variable $b$; none of these variations can be expressed as a combination of others. This is a reasonable assumption because, with multiple diverse environments, it ensures that each domain provides unique information about how $b$ varies, making the latent factors identifiable rather than entangled. The detailed explanations of this lemma can be found in paper Kong et al. (2022a).

**Discussion of Assumptions.** **(A1) Smooth and positive density.** This assumption requires that the conditional densities $p(\mathbf{z} \mid e, y)$ are smooth and strictly positive on their supports. Intuitively, it ensures that the latent space is continuous and that every possible latent configuration has a non-zero probability of occurrence. In practice, this assumption is easily satisfied: with sufficient data, empirical or neural-network-based density estimations naturally produce smooth and positive densities. Since our framework adopts a VAE-based generative model, the latent distribution is modeled by Gaussian families whose densities are positive everywhere, satisfying this assumption automatically. However, this assumption may fail when the latent space is discrete or piecewise constant, such as in deterministic autoencoders or models that collapse the latent variance to zero. In those degenerate cases, $p(\mathbf{z} \mid e, y)$ no longer has a smooth positive density, and the identifiability conditions in Lemma 2.1 no longer hold rigorously.

**(A2) Conditional independence and content invariance.** This assumption states that the coordinates of the latent variables $\mathbf{z} = (z_1, \ldots, z_n)$ are conditionally independent given $e$ and $y$, i.e., $\log p(\mathbf{z} \mid e, y) = \sum_i q_i(z_i; e, y)$. Conceptually, this means that each latent dimension corresponds to an independent generative factor once the environment and label are specified. It also implies that content variables are invariant across environments—only the bias-related coordinates are affected by domain change. This assumption is reasonable in our framework because, when the generative process is disentangled and the noise level is small, the variation of $\mathbf{z}$ can indeed be attributed to independent factors conditioned on $e$ and $y$. Moreover, the VAE objective encourages conditional independence through the factorization of the approximate posterior. However, this assumption may be violated when latent variables interact strongly (e.g., through nonlinear dependencies) or when noise in the data introduces correlated variations between coordinates. In such cases, exact independence cannot be guaranteed, but approximate independence is often sufficient for stable learning and representation disentanglement.

**(A3) Linear independence.** This assumption imposes a technical requirement on the partial derivatives of the latent log-density functions with respect to environment changes. Intuitively, this means that the changes introduced by different environments are sufficiently diverse so that the bias components can be linearly separated from one another. In practice, this condition cannot be enforced directly, but it is often satisfied when multiple environments exist and differ in a sufficiently rich manner. Empirically, in synthetic datasets, when the number of environments is large, linear independence almost always holds, guaranteeing successful disentanglement. In real-world data, this condition may be partially violated because the number of distinct environments if we think this is domain number is limited. Nevertheless, prior works (Kong et al., 2022a; Li et al., 2024a) and our empirical results show that even approximate linear independence is sufficient to achieve robust representations. This is may because the so-called domain in domain generalization are large domain and in this domain that there existing small domains to satisfy the assumptions.

**(A4) Domain variability.** This assumption requires that, for at least one pair of distinct environments $e_i \neq e_j$ and the corresponding class $y$, there exists a measurable subset $A_z \subset \mathcal{Z}$ with non-zero probability that cannot be expressed as a purely content-aligned region $\Omega_c \times B$, such that

$$\int_{A_z} p(\mathbf{z} \mid e_i, y) \, dz \;\neq\; \int_{A_z} p(\mathbf{z} \mid e_j, y) \, dz.$$

This ensures that the conditional latent distribution genuinely varies across domains, capturing environment-dependent factors beyond the content component. Because the mapping $\mathbf{x} = g(\mathbf{z})$ is deterministic and noiseless, $p(\mathbf{x} \mid e, y)$ is the push forward of $p(\mathbf{z} \mid e, y)$; therefore, if $p(\mathbf{z} \mid e_i, y) = p(\mathbf{z} \mid e_j, y)$, we would also have $p(\mathbf{x} \mid e_i, y) = p(\mathbf{x} \mid e_j, y)$. However, this equality does not hold in domain generalization, where environments are explicitly assumed to induce observable shifts. To see this, consider that if the joint distributions were identical across environments,

$$p(\mathbf{x}, y, e_i) = p(\mathbf{x}, y, e_j),$$

then by the chain rule,

$$p(\mathbf{x}, y, e) = p(\mathbf{x} \mid y, e)\, p(y \mid e)\, p(e).$$

Integrating both sides over a measurable subset $A_x$ gives

$$\int_{A_x} p(\mathbf{x} \mid y, e_i)\, p(y \mid e_i)\, dx = \int_{A_x} p(\mathbf{x} \mid y, e_j)\, p(y \mid e_j)\, dx.$$

If $p(\mathbf{x} \mid y, e_i) = p(\mathbf{x} \mid y, e_j)$ held for all $A_x$, then equality would require $p(y \mid e_i) = p(y \mid e_j)$. Yet, in our setting, we explicitly consider label shift, meaning $p(y \mid e)$ varies with $e$; hence, this equality cannot generally hold. Consequently, A4 guarantees genuine domain variability at the level of $p(\mathbf{z} \mid e, y)$, which propagates to $p(\mathbf{x} \mid e, y)$ through the deterministic mapping. Nevertheless, A4 may fail under degenerate or unrealistic conditions, such as:

(i) when environments do not induce any shift in latent factors ($p(\mathbf{z} \mid e_i, y) = p(\mathbf{z} \mid e_j, y)$ for all $i, j$), making domains statistically indistinguishable;

(ii) when the mapping $g$ is stochastic or non-injective, collapsing different latent codes into identical observations, which hides latent variability in the observed space;

(iii) or when label shift is absent ($p(y \mid e_i) = p(y \mid e_j)$), eliminating one key source of distributional difference across domains.

These cases violate the core assumption that domains differ in meaningful, non-content-related ways and thus fall outside our framework.

### B.3 Better predictor with adding bias

*Proof.* Under Assumptions A1–A5, we compare the Bayes-optimal risks for predictors based on $c$ versus $(c, b)$.

**Log-loss.** The Bayes-optimal log-loss risks satisfy

$$R_{\log}\big(q^*(\cdot \mid c)\big) = H(Y \mid C), \qquad R_{\log}\big(q^*(\cdot \mid c, b)\big) = H(Y \mid C, B),$$

where

$$H(Y \mid C) = -\mathbb{E}_C\big[\mathbb{E}_{Y \mid C}[\log p(Y \mid C)]\big], \quad H(Y \mid C, B) = -\mathbb{E}_{C,B}\big[\mathbb{E}_{Y \mid C,B}[\log p(Y \mid C, B)]\big].$$

By A5 there is a positive-measure set of $(c, b)$ on which $p(y \mid c, b) \neq p(y \mid c)$. Hence

$$I(Y; B \mid C) = H(Y \mid C) - H(Y \mid C, B) = \mathbb{E}_{C,B}\big[D_{\mathrm{KL}}(p(y \mid c, b) \,\|\, p(y \mid c))\big] > 0,$$

so $H(Y \mid C, B) < H(Y \mid C)$ and therefore

$$R_{\log}\big(q^*(\cdot \mid c, b)\big) < R_{\log}\big(q^*(\cdot \mid c)\big).$$

**Squared-error loss.** The Bayes-optimal squared-error risks satisfy

$$R_{\mathrm{sq}}\big(f^*(c)\big) = \mathbb{E}\big[\mathrm{Var}(Y \mid C)\big], \qquad R_{\mathrm{sq}}\big(f^*(c, b)\big) = \mathbb{E}\big[\mathrm{Var}(Y \mid C, B)\big].$$

By the law of total variance,

$$\mathrm{Var}(Y \mid C) = \mathbb{E}_B\big[\mathrm{Var}(Y \mid C, B)\big] + \mathrm{Var}_B\big(\mathbb{E}[Y \mid C, B] \mid C\big).$$

Since A5 ensures that $B$ influences the conditional mean of $Y$ given $C$, the second term is strictly positive on a set of positive probability. Thus

$$\mathbb{E}\big[\mathrm{Var}(Y \mid C, B)\big] < \mathbb{E}\big[\mathrm{Var}(Y \mid C)\big],$$

and consequently

$$R_{\mathrm{sq}}\big(f^*(c, b)\big) < R_{\mathrm{sq}}\big(f^*(c)\big).$$

This completes the proof that incorporating $b$ strictly reduces both log-loss and squared-error Bayes risks. $\square$

### B.4 THE DERIVATION OF THE KL OPTIMIZATION

We want to solve the following problem:

$$q^*(y \mid \mathbf{b}) = \underset{q(\cdot \mid \mathbf{b})}{\arg\min}\, \mathbb{E}_{e \sim p(e \mid \mathbf{b})}\Big[D_{\mathrm{KL}}\big(p(y \mid \mathbf{b}, e) \,\|\, q(y \mid \mathbf{b})\big)\Big].$$

Let

$$\mathcal{L}[q(\cdot \mid \mathbf{b})] = \mathbb{E}_{e \sim p(e \mid \mathbf{b})}\Big[D_{\mathrm{KL}}\big(p(y \mid \mathbf{b}, E) \,\|\, q(y \mid \mathbf{b})\big)\Big].$$

By the definition of KL divergence, for discrete $y$:

$$D_{\mathrm{KL}}\big(P(y) \,\|\, Q(y)\big) = \sum_y P(y) \log \frac{P(y)}{Q(y)}.$$

Hence, for each $e$,

$$D_{\mathrm{KL}}\big(p(y \mid \mathbf{b}, e) \,\|\, q(y \mid \mathbf{b})\big) = \sum_y p(y \mid \mathbf{b}, e) \log \frac{p(y \mid \mathbf{b}, e)}{q(y \mid \mathbf{b})}.$$

Therefore,

$$\mathcal{L}[q] = \sum_e p(e \mid \mathbf{b}) \sum_y p(y \mid \mathbf{b}, e) \log \frac{p(y \mid \mathbf{b}, e)}{q(y \mid \mathbf{b})}.$$

We want to choose $q(y \mid \mathbf{b})$ to minimize $\mathcal{L}[q]$, which needs $q(y \mid \mathbf{b})$ to satisfy $\sum_y q(y \mid \mathbf{b}) = 1$, for each fixed $\mathbf{b}$. To incorporate this constraint, we introduce a Lagrange multiplier $\lambda$ and consider the Lagrangian:

$$\mathcal{J}[q, \lambda] = \sum_e p(e \mid \mathbf{b}) \sum_y p(y \mid \mathbf{b}, e) \log\!\left[\frac{p(y \mid \mathbf{b}, e)}{q(y \mid \mathbf{b})}\right] + \lambda\Big(\sum_y q(y \mid \mathbf{b}) - 1\Big).$$

To find the minimizing $q$, we set

$$\frac{\partial \mathcal{J}}{\partial q(y \mid \mathbf{b})} = 0.$$

First, note

$$\log \frac{p(y \mid \mathbf{b}, e)}{q(y \mid \mathbf{b})} = \log p(y \mid \mathbf{b}, e) - \log q(y \mid \mathbf{b}).$$

Only the term $-\log q(y \mid \mathbf{b})$ depends on $q$. For a specific $y$,

$$\frac{\partial}{\partial q(y \mid \mathbf{b})}\big[-\log q(y' \mid \mathbf{b})\big] = \begin{cases} -\frac{1}{q(y \mid \mathbf{b})}, & \text{if } y' = y, \\ 0, & \text{if } y' \neq y. \end{cases}$$

Hence, for each $e$:

$$\frac{\partial}{\partial q(y \mid \mathbf{b})}\left[\sum_{y'} p(y' \mid \mathbf{b}, e) \log \frac{p(y' \mid \mathbf{b}, e)}{q(y' \mid \mathbf{b})}\right] = -\frac{p(y \mid \mathbf{b}, e)}{q(y \mid \mathbf{b})}.$$

Summing over $e$,

$$\frac{\partial}{\partial q(y \mid \mathbf{b})} \sum_e p(e \mid \mathbf{b}) \sum_{y'} p(y' \mid \mathbf{b}, e) \log \frac{p(y' \mid \mathbf{b}, e)}{q(y' \mid \mathbf{b})} = -\frac{1}{q(y \mid \mathbf{b})} \sum_e p(e \mid \mathbf{b})\, p(y \mid \mathbf{b}, e).$$

Also,

$$\frac{\partial}{\partial q(y \mid \mathbf{b})}\Big[\lambda\big(\sum_{y'} q(y' \mid \mathbf{b}) - 1\big)\Big] = \lambda.$$

Hence,

$$\frac{\partial \mathcal{J}}{\partial q(y \mid \mathbf{b})} = -\frac{1}{q(y \mid \mathbf{b})} \sum_e p(e \mid \mathbf{b})\, p(y \mid \mathbf{b}, e) + \lambda.$$

Setting this to zero:

$$-\frac{1}{q(y \mid \mathbf{b})} \sum_e p(e \mid \mathbf{b}) \, p(y \mid \mathbf{b}, e) + \lambda = 0,$$

which gives

$$\frac{1}{q(y \mid \mathbf{b})} \sum_e p(e \mid \mathbf{b}) \, p(y \mid \mathbf{b}, e) = \lambda.$$

Thus

$$q(y \mid \mathbf{b}) = \frac{1}{\lambda} \sum_e p(e \mid \mathbf{b}) \, p(y \mid \mathbf{b}, e).$$

We require $\sum_y q(y \mid \mathbf{b}) = 1$. Then:

$$1 = \sum_y q(y \mid \mathbf{b}) = \sum_y \frac{1}{\lambda} \sum_e p(e \mid \mathbf{b}) \, p(y \mid \mathbf{b}, e) = \frac{1}{\lambda} \sum_e p(e \mid \mathbf{b}) \sum_y p(y \mid \mathbf{b}, e).$$

Since $\sum_y p(y \mid \mathbf{b}, e) = 1$ for each $e$, we get

$$\sum_e p(e \mid \mathbf{b}) \underbrace{\sum_y p(y \mid \mathbf{b}, e)}_{=1} = \sum_e p(e \mid \mathbf{b}) = 1.$$

Hence

$$1 = \frac{1}{\lambda} \cdot 1 \quad \Longrightarrow \quad \lambda = 1.$$

Therefore,

$$q(y \mid \mathbf{b}) = \sum_e p(e \mid \mathbf{b}) \, p(y \mid \mathbf{b}, e).$$

### B.5 DECOMPOSITION AND INVARIANT

Assume by causal representation learning, we have obtained $\mathbf{c}$ (stable latent variables) and $\mathbf{b}$ (unstable latent variables). Our goal is to estimate $p(y \mid \mathbf{c}, \mathbf{b})$.

$$
\begin{aligned}
&\frac{p(y = 1 \mid \mathbf{c}, \mathbf{b})}{p(y = 0 \mid \mathbf{c}, \mathbf{b})} \\
&= \frac{p(\mathbf{c}, \mathbf{b}, y = 1)}{p(\mathbf{c}, \mathbf{b}, y = 0)} \\
&= \frac{p(\mathbf{b} \mid y = 1)p(\mathbf{c} \mid y = 1)p(y = 1)}{p(\mathbf{b} \mid y = 0)p(\mathbf{c} \mid y = 0)p(y = 0)} \\
&= \frac{p(y = 1 \mid \mathbf{b})p(\mathbf{c} \mid y = 1)}{p(y = 0 \mid \mathbf{b})p(\mathbf{c} \mid y = 0)} \\
&= \frac{p(y = 1 \mid \mathbf{b})\frac{p(y=1|\mathbf{c})}{p(y=1)}}{p(y = 0 \mid \mathbf{b})\frac{p(y=0|\mathbf{c})}{p(y=0)}}
\end{aligned}
\tag{14}
$$

Assume we have $e_{all} = \{e_0, e_1, ..., e_n\}$, representing different domains. We assume the source domains are $e_s = \{e_0, e_1, e_s\}$, and target domains are $e_t = \{e_{s+1}, ..., e_n\}$.

$$
\begin{aligned}
&\forall e_i, e_j \in e_{all} \\
&\mathbf{c} \text{ only depends on } y \\
&p(\mathbf{c} \mid y = 1, e = e_i) = p(\mathbf{c} \mid y = 1, e = e_j) \\
&\mathbf{Use \ Bayes \ rule} \\
&\frac{p(y = 1, e = e_i \mid \mathbf{c})}{p(y = 1, e = e_i)} = \frac{p(y = 1, e = e_j \mid \mathbf{c})}{p(y = 1, e = e_j)}
\end{aligned}
\tag{15}
$$

$$p(y = 1 \mid \mathbf{c}) = \sum_{e \in e_s} \frac{p(y = 1, e = e \mid \mathbf{c})}{p(y = 1, e = e)} p(e \mid e_s)$$

This result allows us to estimate $p(y = 1 \mid \mathbf{c})$ using data from source domains and apply it in the target domain.

## B.6 PROOF OF EQUATION (1)

The key observation is that because $\hat{y}$ only depends on $y$ (through $\mathbf{c}$), we can relate the conditional distribution of the pseudo-label $\hat{y}$ given $\mathbf{b}$ to the distribution of the true label $y$. Formally,

$$p(\hat{y} = 1|\mathbf{b}) = p(\hat{y} = 1|y = 1, \mathbf{b})p(y = 1|\mathbf{b}) + p(\hat{y} = 1|y = 0, \mathbf{b})p(y = 0|\mathbf{b})$$

**Consider $\hat{y}$ only depends on** $y, \hat{y} \perp\!\!\!\perp \mathbf{b}|y$

$$= p(\hat{y} = 1|y = 1)p(y = 1|\mathbf{b}) + p(\hat{y} = 1|y = 0)p(y = 0|\mathbf{b})$$
$$= h_1 p(y = 1|\mathbf{b}) + (1 - h_0)(1 - p(y = 1|\mathbf{b}))$$

Solving for $p(y = 1 \mid \mathbf{b})$ gives Equation (1).

## B.7 PROOF AND DISCUSSION OF EQUATION 4.2

**Setup and assumptions.** Write

$$h_0 = p(\hat{y} = 0 \mid y = 0), \qquad h_1 = p(\hat{y} = 1 \mid y = 1), \qquad J = h_0 + h_1 - 1 \in (0, 1].$$

Let $p(y = 1 \mid \mathbf{b})$ denote the target conditional and $q(\mathbf{b}) = p(\hat{y} = 1 \mid \mathbf{b})$. We assume (i) $\mathbf{c} \perp e \mid y$ and $\mathbf{b} \perp \mathbf{c} \mid y$, hence $\hat{y} = f_c(\mathbf{c})$ implies $\hat{y} \perp \mathbf{b} \mid y$; (ii) class-conditional noise (CCN): $p(\hat{y} = 1 \mid y = 1, \mathbf{b}, e) = h_1$ and $p(\hat{y} = 1 \mid y = 0, \mathbf{b}, e) = 1 - h_0$ for all $(\mathbf{b}, e)$; (iii) a margin assumption $p(y = 1 \mid \mathbf{b}) \in [\rho, 1 - \rho]$ almost surely, for some $\rho \in (0, 1/2)$.

**Step 1: Identification under CCN.** By $\hat{y} \perp \mathbf{b} \mid y$ and CCN,

$$q(\mathbf{b}) = p(\hat{y} = 1 \mid \mathbf{b}) = p(\hat{y} = 1 \mid y = 1) p(y = 1 \mid \mathbf{b}) + p(\hat{y} = 1 \mid y = 0) p(y = 0 \mid \mathbf{b})$$
$$= (1 - h_0) + J p(y = 1 \mid \mathbf{b}),$$

hence

$$p(y = 1 \mid \mathbf{b}) = \frac{q(\mathbf{b}) + h_0 - 1}{J}. \tag{16}$$

**Step 2: Linearization of the corrected estimator and $L^2$ bound.** Define the corrected bias predictor

$$\widehat{f}(\mathbf{b}) = \frac{\widehat{q}(\mathbf{b}) + \hat{h}_0 - 1}{\hat{h}_0 + \hat{h}_1 - 1},$$

where $\widehat{q}(\mathbf{b})$ estimates $q(\mathbf{b})$ and $(\hat{h}_0, \hat{h}_1)$ estimate $(h_0, h_1)$ from sources. Consider the map $F(q, h_0, h_1) = (q + h_0 - 1)/(h_0 + h_1 - 1)$. A first-order expansion of $F$ at $(q, h_0, h_1)$ yields, pointwise in $\mathbf{b}$,

$$\widehat{f}(\mathbf{b}) - p(y = 1 \mid \mathbf{b}) = \frac{\widehat{q}(\mathbf{b}) - q(\mathbf{b})}{J} + \frac{1 - p(y = 1 \mid \mathbf{b})}{J} (\hat{h}_0 - h_0) - \frac{p(y = 1 \mid \mathbf{b})}{J} (\hat{h}_1 - h_1) + \text{h.o.t.} \tag{17}$$

(The identities $\partial F/\partial q = 1/J$, $\partial F/\partial h_0 = (1 - p)/J$, and $\partial F/\partial h_1 = -p/J$ follow from equation 16.) Squaring equation 17, using $0 \le p(y = 1 \mid \mathbf{b}) \le 1$, and taking expectation over $\mathbf{b} \sim p(\mathbf{b})$, we obtain

$$\mathbb{E}\big[(\widehat{f}(\mathbf{b}) - p(y = 1 \mid \mathbf{b}))^2\big] \le \frac{2}{J^2} \mathbb{E}\big[(\widehat{q}(\mathbf{b}) - q(\mathbf{b}))^2\big] + \frac{C}{J^2} \Big((\hat{h}_0 - h_0)^2 + (\hat{h}_1 - h_1)^2\Big) + o(1), \tag{18}$$

for an absolute constant $C > 0$ (the $o(1)$ term collects higher-order products of estimation errors).

**Step 3: From $L^2$ error to KL risk.** For any $p, q \in [\rho, 1 - \rho]$,

$$\mathrm{KL}\big(\mathrm{Bern}(p) \,\|\, \mathrm{Bern}(q)\big) \;\leq\; C_\rho\,(p - q)^2, \qquad C_\rho = \frac{1}{\rho(1 - \rho)}. \tag{19}$$

Therefore, with the risk $\mathcal{R}_{(b)}(g) = \mathbb{E}_{\mathbf{b}}[\mathrm{KL}(\mathrm{Bern}(p(y = 1 \mid \mathbf{b})) \,\|\, \mathrm{Bern}(g(\mathbf{b})))]$ and $\mathcal{R}_{(b)}(p) = 0$,

$$\mathcal{R}_{(b)}(\widehat{f}) \leq C_\rho \, \mathbb{E}\big[(\widehat{f}(\mathbf{b}) - p(y = 1 \mid \mathbf{b}))^2\big]$$

$$\overset{equation\ 18}{\leq} \frac{c_1}{J^2} \, \mathbb{E}\big[(\widehat{q}(\mathbf{b}) - q(\mathbf{b}))^2\big] + \frac{c_2}{J^2} \left( (\hat{h}_0 - h_0)^2 + (\hat{h}_1 - h_1)^2 \right),$$

where $c_1 = 2C_\rho$ and $c_2 = C_\rho C$ depend only on the margin parameter $\rho$. This is exactly the claimed upper bound equation 6. $\square$

**Discussion.** The bound in equation 6 shows that the entire target risk of the corrected predictor, $\mathcal{R}_{(b)}(\widehat{f})$, is controlled by a data–driven correction cost: (i) the estimation error $\mathbb{E}\big[(\widehat{q}(\mathbf{b}) - q(\mathbf{b}))^2\big]$ from learning $q(\mathbf{b}) = p(\hat{y} = 1 \mid \mathbf{b})$ on unlabeled target data, and (ii) the squared errors $(\hat{h}_0 - h_0)^2 + (\hat{h}_1 - h_1)^2$ from estimating the pseudo-label noise rates on sources. Both terms are scaled by $(h_0 + h_1 - 1)^{-2}$, so a more informative pseudo-labeler (larger $J = h_0 + h_1 - 1$) yields weaker error amplification. As unlabeled target size $N$ grows and source labels $m$ increase, these components shrink ($\mathbb{E}[(\widehat{q} - q)^2] = O(1/N)$, $(\hat{h}_i - h_i)^2 = O(1/m)$), making the right–hand side small. Consequently, whenever this upper bound is smaller than the source-only risk $\mathcal{R}_{(b)}(f^\star)$ (equivalently, the domain gap), the corrected predictor $\widehat{f}$ is strictly better on the target bias subproblem.

## C  MULTI-CLASS METHOD

Suppose we have $K \geq 2$ classes. We "one-hot encode" these classes, so that $y$ takes values in the set $\mathcal{Y} = \{(1, 0, ..., 0), (0, 1, 0, ..., 0), ..., (0, ..., 0, 1)\} \subseteq \{0, 1\}^K$. In the training stage, let $\hat{y} = f_{\mathbf{c}}(\mathbf{c})$ $\epsilon \in [0, 1]^{\mathcal{Y} \times \mathcal{Y}}$ with

$$\epsilon_{y_i, y_j} = p[\hat{y} = y_i \mid y = y_j]$$

denote the class-conditional confusion matrix of the pseudo-labels. Then, we have

$$
\begin{aligned}
&\frac{p(y = y_i \mid \mathbf{c}, \mathbf{b})}{p(y \neq y_i \mid \mathbf{c}, \mathbf{b})} \\
&= \frac{p(\mathbf{c}, \mathbf{b}, y = y_i)}{\sum_{y_j \neq y_i} p(\mathbf{c}, \mathbf{b}, y = y_j)} \\
&= \frac{p(\mathbf{c}, \mathbf{b} \mid y = y_i)p(y = y_i)}{\sum_{y_j \neq y_i} p(\mathbf{c}, \mathbf{b} \mid y = y_j)p(y = y_j)} \\
&\quad \text{consider } \mathbf{c} \perp\!\!\!\perp \mathbf{b} \mid y : \\
&= \frac{p(\mathbf{c} \mid y = y_i)p(\mathbf{b} \mid y = y_i)p(y = y_i)}{\sum_{y_j \neq y_i} p(\mathbf{c} \mid y = y_j)p(\mathbf{b} \mid y = y_j)p(y = y_j)} \\
&= \frac{p(y = y_i \mid \mathbf{b})p(\mathbf{c} \mid y = y_i)}{\sum_{y_j \neq y_i} p(y = y_j \mid \mathbf{b})p(\mathbf{c} \mid y = y_j)} \\
&= \frac{p(y = y_i \mid \mathbf{b})\frac{p(y = y_i \mid \mathbf{c})}{p(y = y_i)}}{\sum_{y_j \neq y_i} p(y = y_j \mid \mathbf{b})\frac{p(y = y_j \mid \mathbf{c})}{p(y = y_j)}}
\end{aligned}
\tag{20}
$$

for $P \in R^{\mathcal{Y}}$ defined by

$$P_{y_i} = \frac{p(y = y_i \mid \mathbf{b})p(y = y_i \mid \mathbf{c})}{p(y = y_i)} \qquad \text{for each } y_i \in \mathcal{Y}.$$

By using the logit method, we obtain:

$$\text{logit}(p(y = y_i \mid \mathbf{c}, \mathbf{b})) = \log\left(\frac{P_{y_i}}{\sum_{y_j \neq y_i} P_{y_j}}\right)$$

$$= \text{logit}\left(\frac{P_{y_i}}{\|P\|_1}\right) \tag{21}$$

Applying the sigmoid function to each side, we have

$$p(y = y_i \mid \mathbf{c}, \mathbf{b}) = \frac{P}{\|P\|_1}.$$

Like binary classification, we only need to estimate $p(y = y_i \mid \mathbf{b}, e)$ since we assume that $p(y = y_i \mid \mathbf{c})$ and $p(y = y_i)$ can be directly used.

$$E[\hat{y} \mid \mathbf{b}]$$

$$= \sum_{y_i \in \mathcal{Y}} E[\hat{y} \mid y = y_i, \mathbf{b}]p[y = y_i \mid \mathbf{b}]$$

$$\hat{y} \textbf{ only depends on } y \tag{22}$$

$$= \sum_{y_i \in \mathcal{Y}} E[\hat{y} \mid y = y_i]p[y = y_i \mid \mathbf{b}]$$

$$= \epsilon E[y \mid \mathbf{b}]$$

When $\epsilon$ is non-singular, this has the unique solution $E[y \mid \mathbf{b}] = \epsilon^{-1} E[\hat{y} \mid \mathbf{b}]$, giving a multiclass equivalent. However, it is numerically more stable to estimate $E[y \mid \mathbf{b}]$ by the least-squares solution.

$$\text{argmin}_{p \in \Delta^{\mathcal{Y}}} \|\epsilon p - E[\hat{y} \mid \mathbf{b}]\|_2.$$

# D  ALGORITHMS

## D.1  BINARY BIAS-AWARE GENERALIZATION

---
**Algorithm 1** Bias-Aware Generalization

---
1: **Input:** Source data $(\mathbf{x}^{\mathcal{S}_i}, \mathbf{y}^{\mathcal{S}_i})_{i=1}^M$; hyper-parameters $\lambda_0, \lambda_1$,
2: : In the following content $\mathcal{L}$ means loss, $f$ means classier, $e_k$ is the domain embedding, $Pr$ means logits prior, $\text{logit}(p(y))$.
3: **Stage 1: Source Training**
4: Obtain $[\mathbf{c}, \mathbf{b}]$ via feeding $\mathbf{x}$ into encoder.
5: Reconstruct $\hat{\mathbf{x}}$.
6: Compute VAE loss $\mathcal{L}_{\text{vae}}$.
7: Calculate the results of the invariant predictor $f_c(\mathbf{c})$ .
8: Calculate the bias predictor, obtain the final prediction.
9: Compute the classification loss $\mathcal{L}_{\text{cls}}$.
10: Add $\mathcal{L}_{\text{ind}}$ to enforce $\mathbf{c} \perp\!\!\!\perp \mathbf{b} \mid y$.
11: Compute $\mathcal{L}_{\text{all}} = \mathcal{L}_{\text{cls}} + \lambda_0 \mathcal{L}_{\text{vae}} + \lambda_1 \mathcal{L}_{\text{ind}}$ and optimize parameters.
12: For source data, calculate $h_0 = P(\hat{y} = 0 \mid y = 0), h_1 = P(\hat{y} = 1 \mid y = 1)$ using $\hat{y} = f_c(\mathbf{c})$.
13: **Stage 2: Test-Time Adaptation**
14: Obtain $(\mathbf{c}^{\mathcal{T}}, \mathbf{b}^{\mathcal{T}})$ with pre-trained encoders.
15: Generate pseudo-label as $\hat{y}^{\mathcal{T}} = f_c(\mathbf{c}^{\mathcal{T}})$.
16: Fine-tune bias predictor $f_b$ with $(\mathbf{b}^{\mathcal{T}}, \hat{y})$
17: Correct the the results of bias predictor as $\phi(\tilde{f}_b(\mathbf{b}^{\mathcal{T}})) = \text{logit}((\frac{\sigma((\tilde{f}_b(\mathbf{b}^{\mathcal{T}}))) + h_0 - 1}{h_0 + h_1 - 1}))$ ,
18: Make the final prediction.

---

## D.2 MULTI-CLASS BIAS-AWARE GENERALIZATION

---

**Algorithm 2** Multi-Class Bias-Aware Generalization

---

1: **Input:** Source data $(\mathbf{x}^{\mathcal{S}_i}, \mathbf{y}^{\mathcal{S}_i})_{i=1}^{M}$; hyper-parameters $\lambda_0, \lambda_1$,
2: : In the following content $\mathcal{L}$ means loss, $f$ means classier, $e_k$ is the domain embedding
3: **Stage 1: Source Training**
4: Obtain $[\mathbf{c}, \mathbf{b}]$ via feeding $\mathbf{x}$ into encoder.
5: Reconstruct $\hat{\mathbf{x}}$.
6: Compute VAE loss $\mathcal{L}_{\text{vae}}$.
7: Calculate the results of the invariant predictor $f_c(\mathbf{c})$.
8: Calculate the bias predictor, obtain the final prediction as equation 21.
9: Compute the classification loss $\mathcal{L}_{\text{cls}}$.
10: Add $\mathcal{L}_{\text{ind}}$ to enforce $\mathbf{c} \perp\!\!\!\perp \mathbf{b} \mid y$.
11: Compute $\mathcal{L}_{\text{all}} = \mathcal{L}_{\text{cls}} + \lambda_0 \mathcal{L}_{\text{vae}} + \lambda_1 \mathcal{L}_{\text{ind}}$ and optimize parameters.
12: For source data, calculate Probability Confusion Matrix $\epsilon$
13: **Stage 2: Test-Time Adaptation**
14: Obtain $(\mathbf{c}^{\mathcal{T}}, \mathbf{b}^{\mathcal{T}})$ with pre-trained encoders.
15: Generate pseudo-label as $\hat{y}^{\mathcal{T}} = f_c(\mathbf{c}^{\mathcal{T}})$.
16: Fine-tune bias predictor $f_b$ with $(\mathbf{b}^{\mathcal{T}}, \hat{y})$
17: Correct the the results of bias predictor by least squares using $\epsilon$ according Appendix C
18: Make the final prediction as equation 21.

---

# E EXPERIMENT DETAILS AND ABLATION STUDY

## E.1 EXPERIMENT DETAILS

**PACS** We evaluate our BAG model on the PACS dataset, which comprises four distinct domains (Photo, Art painting, Cartoon, Sketch) and seven object categories. Following the standard leave-one-domain-out protocol, we train on three domains and test on the held-out domain. Our backbone is a ResNet-18 pretrained on ImageNet, used to extract visual features. On top of this, we attach a variational autoencoder that separates causal factors from spurious factors, and we introduce learnable domain embeddings to encourage independence across environments. All key hyperparameters—such as the dimensions of the latent representations, the relative weighting between reconstruction and regularization losses, and the learning rates—are chosen via Bayesian optimization, with test-domain accuracy serving as the selection criterion. We train with the Adam optimizer for 70 epochs and select the final model based on its performance under test-time augmentation on the held-out domain. We used the code based on Jiang & Veitch (2022).

**Office-Home** We evaluate our BAG model on the Office-Home dataset Venkateswara et al. (2017), which comprises four domains (Art, Clipart, Product, Real-World) and 65 object categories. Following the leave-one-domain-out protocol, we train on three domains and test on the held-out one. Hyperparameters for each held-out domain are optimized via Bayesian optimization; the selected values for learning rate, regularization weight, number of epochs for the $z_s$ classifier, and VAE weight are given in Table 4. The full code are provided in the supplementary material.

Table 4: Optimal hyperparameters on Office-Home

| Domain | Learning Rate | $\lambda_1$ | # epochs (test time) | $\lambda_0$ |
|---|---|---|---|---|
| Art (Ar) | $5.7 \times 10^{-5}$ | 0.030 | 3 | $1.2 \times 10^{-4}$ |
| Clipart (Cl) | $6.6 \times 10^{-5}$ | 0.012 | 3 | $2.1 \times 10^{-5}$ |
| Product (Pr) | $5.1 \times 10^{-5}$ | 0.030 | 4 | $2.2 \times 10^{-4}$ |
| Real-World (Rw) | $5.0 \times 10^{-5}$ | 0.046 | 6 | $2.5 \times 10^{-5}$ |

Table 5: Accuracy of different variants including **BAG**, BAG-VAE, BAG-RE, and BAG-TTA on four PACS domains and Synthetic data.

| Algorithm | PACS | | | | | Synthetic |
|---|---|---|---|---|---|---|
| | P | A | C | S | Avg | Acc |
| **BAG** | 96.6 | 86.0 | 77.9 | 73.0 | 83.4 | 97.7 |
| **BAG-VAE** | 95.8 | 84.5 | 78.9 | 66.8 | 81.5 | 94.5 |
| **BAG-RE** | 96.1 | 82.8 | 78.4 | 69.7 | 81.8 | 77.0 |
| **BAG-TTA** | 96.3 | 80.0 | 78.8 | 73.0 | 82.0 | 85.5 |

Table 6: Different parts performance results on PACS. Values are accuracies (%).

| Method | P | A | C | S | Avg |
|---|---|---|---|---|---|
| Content predictor Only | 95.57 | 81.10 | 75.00 | 67.83 | 79.88 |
| Bias predictor Only | 94.01 | 73.58 | 78.67 | 66.23 | 78.12 |
| BAG | 96.29 | 83.69 | 80.29 | 76.05 | 84.08 |

## E.2 ABLATION STUDY

Table 5 presents a systematic ablation of BAG on PACS and in our synthetic simulations. The full method comprises two optimization stages and a VAE module. In the ablations, we remove exactly one component at a time, as indicated by a hyphen in the variant name: **BAG-VAE** drops the **VAE** in favor of a linear projection; **BAG–TTA** removes the **T**est **T**ime **A**daptation that uses pseudo labels to guide the bias branch; and **BAG–RE** removes the **R**eweight **E**xperents mechanism. For transparency on the predictive roles of the two branches, Table 6 further reports the behavior of the bias predictor and the content predictor when used in isolation.

Empirically, all BAG variants outperform ERM across datasets, confirming the effectiveness of our bias–utilization design. BAG–RE and BAG–TTA show complementary strengths and weaknesses—consistent with data–specific factors such as domain composition and the severity of distribution shift—whereas BAG–VAE lags behind the full BAG, underscoring the value of accurately identifying and disentangling $\mathbf{b}$ (bias–aligned factors) and $\mathbf{c}$ (content–aligned factors). Using only the bias predictor or only the content predictor yields average accuracies of roughly 78% and 80%, respectively; each alone lacks sufficient information and, without a learnable class prior, is vulnerable to label shift. Once the two predictors are combined under a learnable label prior, performance rises to state–of–the–art levels among the baselines considered.

In summary, the ablations show that each component contributes meaningfully: (i) explicit disentanglement of bias and content, (ii) expert reweighting and test–time adaptation to stabilize routing, and (iii) a compact generative prior via the VAE. Together, these elements enable BAG to deliver robust and consistently strong domain–generalization performance, while simplified variants remain competitive and illuminate the role of each design choice.

## E.3 SENSITIVITY OF EXPERTS AND EMBEDDINGS

Table 7: Sensitivity with fixed experts $E=3$ and varying embedding dimension $d$. Column-wise best in **bold**. Values are accuracies (%).

| $Ds$ | $P$ | $A$ | $C$ | $S$ | average |
|---|---|---|---|---|---|
| 5 | 95.81 | 84.03 | 78.71 | 70.73 | 82.32 |
| 10 | 96.29 | 83.69 | **80.29** | **76.05** | **84.08** |
| 15 | **96.53** | **84.08** | 69.24 | 71.54 | 80.35 |
| 20 | 96.23 | 83.30 | 75.85 | 76.00 | 82.85 |

**Is the environmental predictor necessary, or could one directly predict the label with similar performance?** Yes, the environmental predictor is necessary. Directly predicting $y$ from bias–aligned

Table 8: Sensitivity with fixed embedding dimension $d=10$ and varying number of experts $E$. Column-wise best in **bold**. Values are accuracies (%).

| $E$ | $P$ | $A$ | $C$ | $S$ | average |
|---|---|---|---|---|---|
| 1 | 95.51 | 83.11 | 75.00 | 74.65 | 82.07 |
| 3 | 96.29 | 83.69 | **80.29** | **76.05** | **84.08** |
| 4 | 95.27 | 83.59 | 77.52 | 68.92 | 81.33 |
| 5 | 95.51 | 84.28 | 75.13 | 73.15 | 82.02 |
| 6 | 96.23 | **85.06** | 73.98 | 73.91 | 82.29 |
| 7 | 95.45 | 84.08 | 74.62 | 73.63 | 81.94 |
| 10 | **96.35** | 83.98 | 78.92 | 73.63 | 83.22 |

features $b$ effectively fits $p(y \mid b)$, which is brittle under label shift: when $p(y)$ changes across domains, the conditional $p(y \mid b)$ drifts, and target performance degrades. Our approach explicitly estimates a latent environment and routes each example to domain–conditioned experts, separating environment–specific variation from task–relevant signals and improving robustness. As shown in Table 5, removing the environmental predictor (method **BAG–RE**) reduces the mean accuracy to 81.8%, below that of the full model. In our hyperparameter sweeps (Tables 7 and 8), most configurations that include the predictor exceed 81.8% on average, and the best setting with $E=3, d=10$ reaches 84.08% with strong performance on the challenging domains $C=80.29\%$ and $S=76.05\%$. In summary, conditioning the classifier on an explicitly predicted environment addresses distribution shift and yields measurable, domain–wise robust gains over direct prediction.

**How sensitive is performance to the number of experts and the embedding dimensionality?** Sensitivity is non–monotonic in both factors. With experts fixed at $E=3$ (Table 7), a medium environment embedding performs best: $d=10$ attains the highest average 84.08%, whereas a smaller embedding underfits ($d=5: 82.32\%$). With embedding fixed at $d=10$ (Table 8), increasing experts from $E=1$ to $E=3$ improves the average from 82.07% to **84.08**%; further increases provide no consistent gains (e.g., $E=10: 83.22\%$). A few well–specialized experts ($E\approx3$) together with a compact environment embedding ($d\approx10$) strike a reliable balance between expressivity and out–of–domain generalization. To summarize: BAG is not sensitive to these two parameters, and the performance is within a controllable range; even suboptimal parameter choices do not push it below baselines such as ERM, IRM and ACTIR.

### E.4 DISENTANGLEMENT OF LATENT VARIABLES

We qualitatively examine the learned disentanglement on PACS by visualizing attributions for the content branch **c** and the bias branch **b** with Grad-CAM. Concretely, we reuse the trained checkpoint and the identical backbone as training. Given an input image $x$, we route it through the original BAG model but expose two thin wrappers that return the logits of the **c** head ($f_\beta$) and the **b** head ($f_\eta$), respectively. For Grad-CAM, we target the last residual block of `layer4` in the backbone. We then compute two grayscale attribution maps: one for **c** by backpropagating through $f_\beta$, and one for **b** by backpropagating through $f_\eta$, each with `ClassifierOutputTarget` set to the ground-truth class.

To make the overlays visually comparable across images, we (i) bilinearly resize each Grad-CAM map to the input resolution; (ii) min–max normalize it to $[0, 1]$; and (iii) render red intensity for **c** and blue intensity for **b** before alpha-blending with the RGB image (default $\alpha = 0.5$). For each sample we compose a four-tile horizontal panel in the fixed order: original image, **b** (blue), **c** (red), and the composite overlay; the titles b/c are boldfaced to reduce ambiguity.

Across the shown examples, **c** consistently concentrates on class-relevant object regions, whereas **b** emphasizes background textures, styles, or peripheral cues. These observations align with the intended decomposition and support the use of **c** as content and **b** as environment/style bias in subsequent modules.

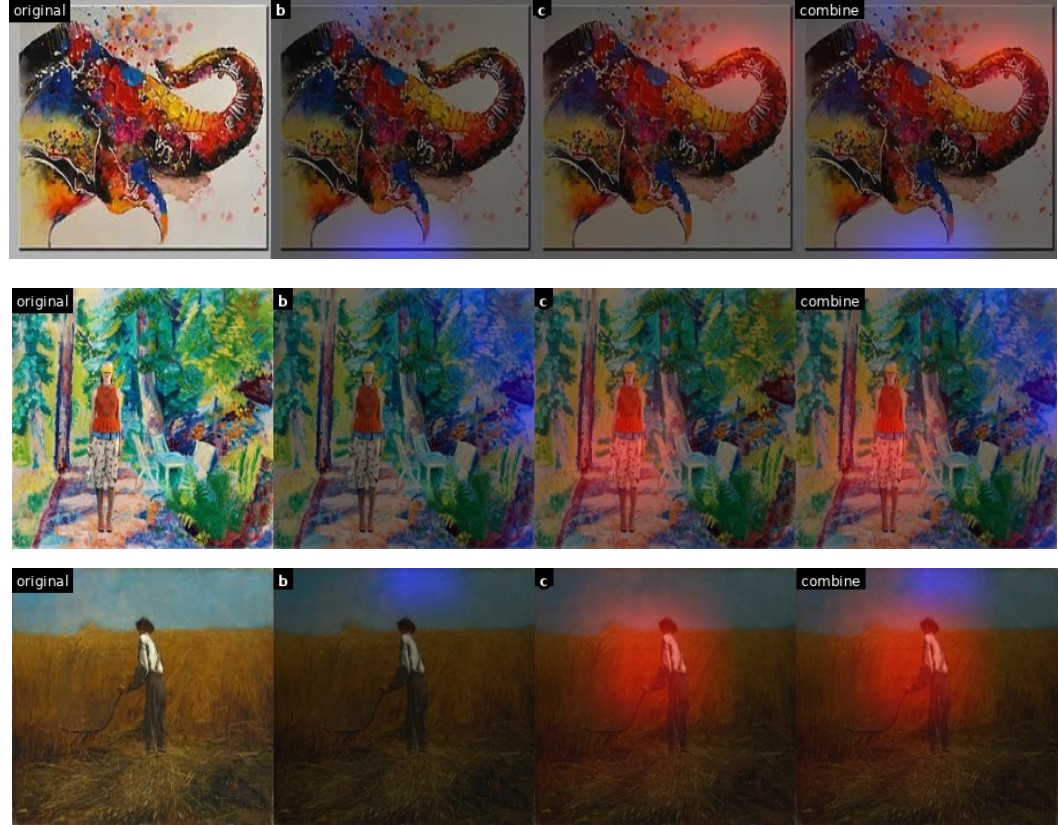

Figure 4: **PACS disentanglement via Grad-CAM.** For each example, we show (left→right): original image; **b** heatmap (blue) obtained by targeting the bias head logits; **c** heatmap (red) obtained by targeting the content head logits; and the composite overlay (intensity reflects attribution strength).The figure displays representative, high-separation cases where **c** focuses on object pixels while **b** emphasizes background/style cues, evidencing effective content–bias separation.

## F  STATEMENT

This work contributes to the advancement of machine learning by challenging conventional out-of-distribution (OOD) approaches and introducing a framework that strategically leverages bias rather than eliminating it. On the positive side, our methods can improve model robustness when deployed in naturally biased environments—e.g., medical imaging systems that must generalize across hospitals with different demographic distributions. By revealing how bias can be harnessed as a constructive signal, we hope to inspire new lines of research into principled bias-aware generalization across diverse domains. Because this is foundational research evaluated on controlled benchmarks and synthetic datasets, we do not identify any immediate negative societal impacts.

