# OpenReview forum: "Should Bias be Eliminated? A General Framework to Use Bias for OOD Generalization"
_ICLR.cc/2026/Conference — Submitted to ICLR 2026_

### Official Review · Reviewer_Gn2a · 2025-10-31

**Soundness:** 3
**Presentation:** 3
**Contribution:** 3
**Rating:** 4
**Confidence:** 3

**Summary:**

This paper questions the conventional treatment of bias in out-of-distribution (OOD) generalization, proposing instead that bias can be constructively leveraged for improved performance. The authors provide a theoretical framework for identifying when and how biased features contribute positively and introduce a generative model-based approach to disentangle invariant content from bias. They further propose a bias-aware prediction scheme that incorporates environment routing and adaptive bias correction with learnable label priors. Extensive experiments on synthetic data and established domain generalization benchmarks demonstrate that their approach (BAG) achieves robust improvements over both invariance-only and prior bias-utilization methods.

**Strengths:**

1.	The paper provides a strong theoretical analysis of the role of bias in ODD generalization, which inspires the design of the proposed framework BAG.
2.	Experiments on two benchmarks prove the effectiveness of the proposed method.
3.	The main ideas, motivations, and methodological details are generally well-presented.

**Weaknesses:**

1.	Theoretical validation is insufficient. Beyond performance results, the theoretical conclusions in the paper lack quantitative experimental support. I recommend adding empirical analyses that directly substantiate the theoretical claims, which would also help reinforce the motivation of the proposed method.
2.	Missing comparison with SOTA DG methods, like [1,2,3]. In addition, I am curious whether the proposed methods can be effectively applied on top of SOTA DG methods.
3.	The empirical results are only demonstrated on relatively modest-scale benchmarks(PACS and OfficeHome). Experiments on larger datasets (like DomainNet or TerraIncognita) should also be provided.

[1] Zou, Yingtian, et al. "Towards robust out-of-distribution generalization bounds via sharpness." arXiv preprint arXiv:2403.06392 (2024).

[2] Zhang, Qingyang, et al. "The best of both worlds: On the dilemma of out-of-distribution detection." Advances in Neural Information Processing Systems 37 (2024): 69716-69746.

[3] Chen, Jinggang, et al. "Gaia: Delving into gradient-based attribution abnormality for out-of-distribution detection." Advances in Neural Information Processing Systems 36 (2023): 79946-79958.

**Questions:**

No supplementary material that provides code, which is mentioned in L1255 that “The full code are provided in the supplementary material”.

**Details Of Ethics Concerns:**

N/A.

---

> ### Author Response · Authors · 2025-11-22
> **Overall reply and reply to weakness 1**
>
> Dear Reviewer Gn2a,
>
> We sincerely appreciate your thoughtful review, recognition of our experiments and presentations, and constructive feedback. We have read the papers you listed and have tried to add more experiments. Here we will answer your concerns and questions one by one as follows:
>
> > W1.Theoretical validation is insufficient. Beyond performance results, the theoretical conclusions in the paper lack quantitative experimental support. I recommend adding empirical analyses that directly substantiate the theoretical claims, which would also help reinforce the motivation of the proposed method.
>
> Thank you for the helpful suggestions, which strengthen the credibility of our conclusions. Following this advice, we conducted a series of ablation studies in simulation to further substantiate our theory. We first describe the data generation pipeline and the model architectures used in the experiments. We sample a discrete environment variable $e \sim \mathrm{Categorical}({\pi_1,\ldots,\pi_M})$ and let $E$ denote the embedding of $e$. We then generate the ground truth binary label
> $y = \mathbf{1}{ w^\top E + b_0 > 0 }.$ Conditional on $y$, we generate the stable content $c$ as $c=c_0+\varepsilon$ if $y=0$ and $c=c_1+\varepsilon$ if $y=1$, with $\varepsilon \sim \mathcal{N}(0,\sigma_c^2 I)$. Next, we construct the bias variable $b$ as $b = E + C_{e,y}$. Finally, $b$ and $c$ jointly determine the input $x$. For most tasks we use a simple VAE whose encoder and decoder are single linear layers. For validating blockwise identification, we adopt iMSDA as the backbone and increase the noise level in the data generation process to better approximate real world conditions.
>
> To verify that blockwise identification of $c$ and $b$ can be satisfied, we use iMSDA as the backbone and vary the number of domains to be 1, 2, and 3. The results are:
>
> | Domain Number |   1   |   2   |   3   |
> | :-----------: | :---: | :---: | :---: |
> |     $R^2$     | 0.441 | 0.575 | 0.738 |
>
> When the number of domains equals $n * b + 1$, where $n * b$ is the dimensionality of the bias variable $b$, the linear independence condition in Assumption A3 is satisfied. In our setup this corresponds to three domains, and $R^2$ reaches a high level. This supports that the bias variable and the stable content meet the requirements of blockwise identification.
>
> To examine which theoretical factors drive BAG performance, we create several variants of the synthetic generator and use our simple VAE based BAG as the common baseline.
>
> i) We enforce a linear relationship between $c$ and $b$ by setting$z_c = \mathrm{center}(Y) + A_{cs} z_s + \epsilon, \quad \epsilon \sim \mathcal{N}(0,\sigma_{z_c}^2 I).$In this regime, $b$ exerts an unblocked influence on $y$, which satisfies Lemma 2.3 and introduces additional signal, so performance improves.
>
> ii) To assess the importance of information in $P(e \mid b)$ for BAG, we modify the generation of $b$ so that it combines the stable content $C$ with noise $\epsilon \sim \mathcal{N}(0,\sigma^2 I)$ instead of combining $C$ with the environment embedding $E$. This substantially reduces the information carried by $P(e \mid b)$ and leads to a marked drop in performance, indicating that environment information encoded in $b$ is crucial for BAG.
>
> iii) To make blockwise identification hard to satisfy, we double the noise strength in the mapping from ${b,c}$ to $x$. Performance degrades notably, which shows that having data distributions that satisfy blockwise identification is also essential for BAG. The ablation results are:
>
> | Ablation               | Acc   |
> | ---------------------- | ----- |
> | BAG                    | 97.48 |
> | Wrong $P(e \mid b)$    | 63.00 |
> | $z_c$ related to $z_s$ | 99.52 |
> | Large $z \to x$ noise  | 66.40 |
>
> We further assess how the number of training domains affects the baselines. We train BAG, ERM, and IRM using only 1, 2, or 3 domains, keeping all other settings fixed. The results are:
>
>
> | Domain Num   | 1     | 2     | 3     |
> | ------------ | ----- | ----- | ----- |
> | BAG（Acc %） | 63.00 | 89.54 | 97.48 |
> | ERM（Acc %） | 57.58 | 58.58 | 54.62 |
> | IRM（Acc %） | 51.18 | 52.82 | 54.73 |
>
> When the number of domains is below the requirement in Assumption A3 of Lemma 2.1, the performance of BAG declines steadily, while the number of domains has no significant impact on ERM and IRM under the same settings. Collectively, these experiments validate the theoretical claims and show that both the presence of environment information in $b$ and the satisfaction of blockwise identification are key determinants of BAG performance, and that the requirement of $n_b + 1$ environments is necessary to realize the expected gains under Assumption A3.

---

> > ### Author Response · Authors · 2025-11-22
> > **Replies to weakness 2 and weakness 3**
> >
> > > W2 .Missing comparison with SOTA DG methods, like [1,2,3]. In addition, I am curious whether the proposed methods can be effectively applied on top of SOTA DG methods.
> > >
> > > [1] Zou, Yingtian, et al. "Towards robust out-of-distribution generalization bounds via sharpness." arXiv preprint arXiv:2403.06392 (2024).
> > >
> > > [2] Zhang, Qingyang, et al. "The best of both worlds: On the dilemma of out-of-distribution detection." Advances in Neural Information Processing Systems 37 (2024): 69716-69746.
> > >
> > > [3] Chen, Jinggang, et al. "Gaia: Delving into gradient-based attribution abnormality for out-of-distribution detection." Advances in Neural Information Processing Systems 36 (2023): 79946-79958.
> >
> > Thank you for pointing out these papers. Paper [1] has no results on the PACS or Office-Home datasets, so we cannot compare it, and it does not provide code. We’ve emailed the authors for the code and would like to conduct experiments if we can get it. Papers [2] and [3] are mainly about out-of-distribution detection, which is different from domain generalization. Out-of-distribution detection decides whether a test sample belongs to the training distribution (ID) or not (OOD), often enabling rejection, while domain generalization aims to maintain high task performance on unseen but label-consistent domains. Therefore, we cannot compare them within our framework. We’ve added these papers to the related works in Line 813, and if we obtain new results, we will immediately add them to the main table.
> >
> > * In addition, I am curious whether the proposed methods can be effectively applied on top of SOTA DG methods
> >
> >   Our method can contribute to SOTA DG methods because we propose that bias, which is commonly ignored, can contribute to DG. We use information in bias that many methods do not exploit, and it would intuitively contribute to improved results. Since we still don't have code of paper[1], we currently are applying our methods over GMDG[11] which is an advanced method recently.  We will report results immediately once we finished it.
> >
> > > W3. The empirical results are only demonstrated on relatively modest-scale benchmarks(PACS and OfficeHome). Experiments on larger datasets (like DomainNet or TerraIncognita) should also be provided.
> >
> > Thank you for your suggestion and the experiments in DomainNet is improtant.  We have added experiments on DomainNet[4], which is a large-scale dataset. We compared BAG against baselines like ERM, MMD[5], IRM[6], ACTIR[7], GMDG[8],MIRO[9], RDM[10], DANN[11], CDANN[12], SagNet[13], EQRM[14]. You can find them in Table 3 in Section 5.2 in the revised version.
> >
> > **DomainNet target-domain accuracies (%) — ResNet-50 backbone** *Most baseline results are taken from [14].*
> >
> > | Algorithm  | Year |  clipart | infograph | painting | quickdraw |     real |   sketch |      Avg |
> > | ---------- | :--: | -------: | --------: | -------: | --------: | -------: | -------: | -------: |
> > | MIRO[9]    | 2022 |       -- |        -- |       -- |        -- |       -- |       -- |     44.3 |
> > | RDM[10]    | 2024 |       -- |        -- |       -- |        -- |       -- |       -- |     43.4 |
> > | ERM        |  —   |     58.1 |      18.8 |     46.7 |      12.2 |     59.6 |     49.8 |     40.9 |
> > | DANN[11]   | 2016 |     53.1 |      18.3 |     44.2 |      11.8 |     55.5 |     46.8 |     38.3 |
> > | MMD[5]     | 2018 |     32.1 |      11.0 |     26.8 |       8.7 |     32.7 |     28.9 |     23.4 |
> > | CDANN[12]  | 2018 |     54.6 |      17.3 |     43.7 |      12.1 |     56.2 |     45.9 |     38.3 |
> > | IRM[6]     | 2019 |     48.5 |      15.0 |     38.3 |      10.9 |     48.2 |     42.3 |     33.9 |
> > | SagNet[13] | 2021 |     57.7 |      19.0 |     45.3 |      12.7 |     58.1 |     48.8 |     40.3 |
> > | EQRM[14]   | 2022 |     56.1 |      19.6 |     46.3 |      12.9 |     61.1 |     50.3 |     41.0 |
> > | ACTIR[7]   | 2022 |     50.0 |      22.6 |     43.5 |      11.7 |     57.8 |     46.8 |     38.7 |
> > | GMDG[8]    | 2024 | **63.4** |      22.4 |     51.4 |      13.4 |     64.4 | **52.4** |     44.6 |
> > | **BAG**    | 2025 |     61.8 |  **25.6** | **51.8** |  **13.9** | **65.5** |     50.3 | **44.8** |

---

> > > ### Author Response · Authors · 2025-11-22
> > > **Replies to weakness 3 and question 1**
> > >
> > > We use a ResNet-50 backbone for feature extraction to match the baselines that also adopt ResNet-50. For MIRO [9] and RDM [10], we report only the mean accuracy because their papers provide averages without per-domain breakdowns. The ACTIR [7] result is obtained by running the authors’ public code under their official setting, as ACTIR does not report DomainNet results. On DomainNet, our method attains the best average accuracy (44.8%), surpassing recent works like GMDG [8] (44.6%) and MIRO [9] (44.3%). At the per-domain level, it is best on infograph, painting, quickdraw, and real, while remaining competitive on clipart and sketch. The largest gain occurs on infograph, likely because background/stylistic artifacts are prominent there; explicitly disentangling content from bias both mitigates spurious domain cues and enables us to leverage bias when it is informative. On clipart and sketch we are not SOTA, which is likely because the domain-specific bias signal in these two domains is weak, uninformative, or misaligned, limiting the benefit of leveraging it. Compared with ACTIR [7] under the same setting, our method achieves substantially higher averages, which we attribute to more accurate disentanglement of, and principled use of, bias.
> > >
> > > [4] Peng X, Bai Q, Xia X, et al. Moment matching for multi-source domain adaptation[C]//Proceedings of the IEEE/CVF International Conference on Computer Vision. 2019: 1406–1415.
> > >
> > > [5] Li H, Pan S J, Wang S, et al. Domain generalization with adversarial feature learning[C]//Proceedings of the IEEE Conference on Computer Vision and Pattern Recognition. 2018: 5400–5409.
> > >
> > > [6] Arjovsky M, Bottou L, Gulrajani I, et al. Invariant risk minimization[J]. arXiv preprint arXiv:1907.02893, 2019.
> > >
> > > [7] Jiang Y, Veitch V. Invariant and transportable representations for anti-causal domain shifts[J]. Advances in Neural Information Processing Systems, 2022, 35: 20782–20794.
> > >
> > > [8] Tan Z, Yang X, Huang K. Rethinking multi-domain generalization with a general learning objective[C]//Proceedings of the IEEE/CVF Conference on Computer Vision and Pattern Recognition. 2024: 23512–23522.
> > >
> > > [9] Cha J, Lee K, Park S, et al. Domain generalization by mutual-information regularization with pre-trained models[C]//European Conference on Computer Vision. Cham: Springer Nature Switzerland, 2022: 440–457.
> > >
> > > [10] Nguyen T, Do K, Duong B, et al. Domain generalisation via risk distribution matching[C]//Proceedings of the IEEE/CVF Winter Conference on Applications of Computer Vision. 2024: 2790–2799.
> > >
> > > [11] Ganin Y, Ustinova E, Ajakan H, et al. Domain-adversarial training of neural networks[J]. Journal of Machine Learning Research, 2016, 17(59): 1–35.
> > >
> > > [12] Long M, Cao Z, Wang J, et al. Conditional adversarial domain adaptation[J]. Advances in Neural Information Processing Systems, 2018, 31.
> > >
> > > [13] Nam H, Lee H J, Park J, et al. Reducing domain gap by reducing style bias[C]//Proceedings of the IEEE/CVF Conference on Computer Vision and Pattern Recognition. 2021: 8690–8699.
> > >
> > > [14] Eastwood C, Robey A, Singh S, et al. Probable domain generalization via quantile risk minimization[J]. Advances in Neural Information Processing Systems, 2022, 35: 17340–17358.
> > >
> > > >Q1 No supplementary material that provides code, which is mentioned in L1255 that “The full code are provided in the supplementary material”.
> > >
> > > Sorry for the oversight. We have attached the code in the supplementary materials. Thank you again for your careful check.

---

### Official Review · Reviewer_Z9H1 · 2025-11-01

**Soundness:** 3
**Presentation:** 3
**Contribution:** 2
**Rating:** 6
**Confidence:** 2

**Summary:**

The paper asks whether “bias” should always be removed for out-of-distribution (OOD) generalization and proposes BAG (Bias-Aware Generalization): (1) learn disentangled content/bias features via a VAE; (2) build a bias-aware predictor using an environment-routing mixture of experts with a learned domain estimator; and (3) perform bias correction with an adaptive label prior, yielding an “invariant predictor” term that remains valid under label shift. Experiments on synthetic data, PACS, and Office-Home show gains over ERM/IRM/ACTIR and recent bias-utilization SFB.

**Strengths:**

1. The paper formalizes when bias can help rather than hurt, via identifiability and “unblocked influence” conditions, then operationalizes it in BAG.
2. The adaptive label prior and decomposition (Eq. 5) address a key weakness of prior SFB-like methods that assume invariant

**Weaknesses:**

1. Conditions A1–A4 (smooth densities, conditional independence factorization, domain variability/linearly independent shifts) may be hard to meet in complex vision tasks; practical diagnostics are not provided.
2. While block-wise identifiability is cited from prior work, the paper’s empirical section offers limited stress-tests for partial violations (e.g., non-identifiable generators, entangled factors)

**Questions:**

1. How sensitive is performance to the number of domain experts and embedding dimensionality? Any heuristics or validation procedure beyond source-domain accuracy?
2. If p(e∣b) is wrong (e.g., target domain outside the convex hull of sources), how does BAG degrade? Any fallback to invariant-only predictions?

---

> ### Author Response · Authors · 2025-11-22
> **Overall reply and reply to weakness 1**
>
> Dear Reviewer Z9H1,
>
> We sincerely appreciate your encouraging comments on our method. Your valuable suggestions for enhancing the theoretical analysis and necessary discussion are highly valued by us. In response to your suggestions, we have added more discussions and experiments. The one-to-one responses can be found below:
>
> > W1 Conditions A1–A4 (smooth densities, conditional independence factorization, domain variability/linearly independent shifts) may be hard to meet in complex vision tasks; practical diagnostics are not provided.
>
> Thank you for the concern of these conditionds and it will help us to discuss about it. We would like to highlight that these assumptions are common in causal representation learning frameworks like in [1] [2] [3] [4]. Some conditions are easy to satisfied in the real world tasks like A4.  A1 and A2 can be applied in the model.  We have already provided detailed discussions of the practical implications and real-world validity of these assumptions in Appendix B.2. We show the relevant content below.
>
>
> **(A1) Smooth and positive density.**
> This assumption requires that the conditional densities $p(\textbf{z} \mid e,y)$ are smooth and strictly positive on their supports. Intuitively, it ensures that the latent space is continuous and that every possible latent configuration has a non-zero probability of occurrence. In practice, this assumption is easily satisfied: with sufficient data, empirical or neural-network-based density estimations naturally produce smooth and positive densities. Since our framework adopts a VAE-based generative model, the latent distribution is modeled by Gaussian families whose densities are positive everywhere, satisfying this assumption automatically. However, this assumption may fail when the latent space is discrete or piecewise constant, such as in deterministic autoencoders or models that collapse the latent variance to zero. In those degenerate cases, $p(\textbf{z} \mid e,y)$ no longer has a smooth positive density, and the identifiability conditions in Lemma 2.1 no longer hold rigorously.
>
> **(A2) Conditional independence and content invariance.**
> This assumption states that the coordinates of the latent variables $\textbf{z} = (z_1,\dots,z_n)$ are conditionally independent given $e$ and $y$, i.e., $\log p(\textbf{z} \mid e,y) = \sum_i q_i(z_i; e,y)$. Conceptually, this means that each latent dimension corresponds to an independent generative factor once the environment and label are specified. It also implies that content variables are invariant across environments—only the bias-related coordinates are affected by domain change. This assumption is reasonable in our framework because, when the generative process is disentangled and the noise level is small, the variation of $\textbf{z}$ can indeed be attributed to independent factors conditioned on $e$ and $y$. Moreover, the VAE objective encourages conditional independence through the factorization of the approximate posterior. However, this assumption may be violated when latent variables interact strongly (e.g., through nonlinear dependencies) or when noise in the data introduces correlated variations between coordinates. In such cases, exact independence cannot be guaranteed, but approximate independence is often sufficient for stable learning and representation disentanglement.
>
> **(A3) Linear independence.**
> This assumption imposes a technical requirement on the partial derivatives of the latent log-density functions with respect to environment changes. Intuitively, this means that the changes introduced by different environments are sufficiently diverse so that the bias components can be linearly separated from one another. In practice, this condition cannot be enforced directly, but it is often satisfied when multiple environments exist and differ in a sufficiently rich manner. Empirically, in synthetic datasets, when the number of environments is large, linear independence almost always holds, guaranteeing successful disentanglement. In real-world data, this condition may be partially violated because the number of distinct environments (if we think this is domain number) is limited. Nevertheless, prior works and our empirical results show that even approximate linear independence is sufficient to achieve robust representations. This may be because the so-called domains in domain generalization are large domains, and within each there exist smaller sub-domains that effectively satisfy the assumptions.

---

> > ### Author Response · Authors · 2025-11-22
> > **Replies to weakness 1 and wekaness 2**
> >
> > **(A4) Domain variability.**
> > This assumption requires that, for at least one pair of distinct environments $e_i \neq e_j$ and the corresponding class $y$, there exists a measurable subset $A_z \subset \mathcal{Z}$ with non-zero probability that cannot be expressed as a purely content-aligned region $\Omega_c \times B$, such that
> > $\int_{A_z} p(\textbf{z} \mid e_i, y)\,dz \;\neq\; \int_{A_z} p(\textbf{z} \mid e_j, y)\,dz .$ This ensures that the conditional latent distribution genuinely varies across domains, capturing environment-dependent factors beyond the content component. Because the mapping $\textbf{x} = g(\textbf{z})$ is deterministic and noiseless, $p(\textbf{x} \mid e,y)$ is the pushforward of $p(\textbf{z} \mid e,y)$; therefore, if $p(\textbf{z} \mid e_i, y)=p(\textbf{z} \mid e_j, y)$, we would also have $p(\textbf{x} \mid e_i, y)=p(\textbf{x} \mid e_j, y)$. However, this equality does not hold in domain generalization, where environments are explicitly assumed to induce observable shifts. To see this, consider that if the joint distributions were identical across environments,$p(\textbf{x}, y, e_i)=p(\textbf{x}, y, e_j)$, then by the chain rule,$p(\textbf{x}, y, e)=p(\textbf{x} \mid y,e)\,p(y \mid e)\,p(e).$ Integrating both sides over a measurable subset $A_x$ gives $\int_{A_x} p(\textbf{x} \mid y,e_i)\,p(y \mid e_i)\,dx \;=\;\int_{A_x} p(\textbf{x} \mid y,e_j)\,p(y \mid e_j)\,dx .$ If $p(\textbf{x} \mid y,e_i) = p(\textbf{x} \mid y,e_j)$ held for all $A_x$, then equality would require $p(y \mid e_i)=p(y \mid e_j)$. Yet, in our setting, we explicitly consider **label shift**, meaning $p(y \mid e)$ varies with $e$; hence, this equality cannot generally hold. Consequently, A4 guarantees genuine domain variability at the level of $p(\textbf{z} \mid e,y)$, which propagates to $p(\textbf{x} \mid e,y)$ through the deterministic mapping. Nevertheless, A4 may fail under degenerate or unrealistic conditions, such as:
> > (i) when environments do not induce any shift in latent factors ($p(\textbf{z} \mid e_i, y)=p(\textbf{z} \mid e_j, y)$ for all ($i,j$), making domains statistically indistinguishable;
> > (ii) when the mapping $g$ is stochastic or non-injective, collapsing different latent codes into identical observations, which hides latent variability in the observed space;
> > (iii) or when label shift is absent ($p(y\mid e_i)=p(y\mid e_j)$), eliminating one key source of distributional difference across domains.
> > These cases violate the core assumption that domains differ in meaningful, non-content-related ways and thus fall outside our framework.
> >
> > [1]Kong L, Xie S, Yao W, et al. Partial identifiability for domain adaptation[J]. arXiv preprint arXiv:2306.06510, 2023.
> >
> > [2]Yao, Dingling, et al. "Multi-view causal representation learning with partial observability." arXiv preprint arXiv:2311.04056 (2023).
> >
> > [3]Lachapelle et al. "Discovering Latent Causal Variables via Mechanism Sparsity: A New Principle for Nonlinear ICA."
> >
> > [4]Hu et al. "Instrumental variable treatment of nonclassical measurement error models."
> >
> > > W2 While block-wise identifiability is cited from prior work, the paper’s empirical section offers limited stress-tests for partial violations (e.g., non-identifiable generators, entangled factors)
> >
> >
> > Thank you for the suggestion; it substantially improves the completeness of our empirical evaluation. To stress test our method when some assumptions are only partially satisfied, we designed two synthetic experiments. We first describe the base generator. Guided by the causal graph in Figure 2, we sample an environment $e \sim \mathrm{Categorical}(\pi_1,\ldots,\pi_M)$ and form a representation $E$. The ground truth binary label is $y=\mathbb{1}{w^\top E + b_0 > 0}$. Conditional on $y$, the stable content is generated as $c=c_0+\epsilon$ if $y=0$ and $c=c_1+\epsilon$ if $y=1$, with $\epsilon\sim\mathcal{N}(0,\sigma_c^2 I)$. The bias variable is then generated as a function of $e$ and $y$; specifically, $b=E+C_{e,y}$. Finally, the observed input is jointly determined by $c$ and $b$ via
> > $x = M[c \  b]^T + \eta,\qquad \eta\sim \mathcal{N}(0,\sigma_x^2 I).$
> >
> > To probe robustness, we consider two variants of this process. In the non identifiable setting, we increase the observation noise by doubling the standard deviation of $\eta$, which makes the mapping from $(c,b)$ to $x$ less identifiable. In the entangled factors setting, we introduce linear coupling between the latent content and the latent spurious or bias factor: for the content latent we set
> > $z_c=\mathrm{center}(Y)+A_{cs}z_s+\varepsilon,\qquad \varepsilon\sim\mathcal{N}(0,\sigma_{z_c}^2 I),$
> > which entangles $c$ and $s$ (with $z_s$ the latent associated with the bias component). We use a simple VAE backbone whose encoder and decoder are each a single linear layer, and evaluate on the unmodified data and on both variants. Accuracy is reported below.

---

> > > ### Author Response · Authors · 2025-11-22
> > > **Replies to weakness 2 and question 1**
> > >
> > > | Ablation                    | Accuracy (%) |
> > > | --------------------------- | ------------ |
> > > | BAG baseline                | 97.48        |
> > > | Entangled factors           | 99.52        |
> > > | Non identifiable generators | 66.40        |
> > >
> > > These results indicate that weakening the identifiability conditions causes a sharp drop in performance, consistent with the increasing difficulty of recovering $c$ and $b$ from a noisier $x$. By contrast, when latent factors are entangled, performance improves; our hypothesis is that the added coupling imposes constraints between $c$ and $b$, creating a shortcut in which learning one variable suffices to infer the other,and thereby $y$ without fully disentangling both.
> > >
> > > Besides, to examine how the model behaves when training domains are scarce, we designed an additional set of experiments. We retained only one, two, or three domains for the training set while keeping all other settings unchanged, and measured how BAG scales with the number of domains. The results in the table show that when the domain count is below the requirement specified in Lemma 2.1, Assumption A3 on linear independence, the performance of BAG declines. Notably‌, the number of domains has no significant impact on ERM and IRM under the same settings.
> > >
> > >
> > >
> > > | Domain Num   | 1     | 2     | 3     |
> > > | ------------ | ----- | ----- | ----- |
> > > | BAG（Acc %） | 63.00 | 89.54 | 97.48 |
> > > | ERM（Acc %） | 57.58 | 58.58 | 54.62 |
> > > | IRM（Acc %） | 51.18 | 52.82 | 54.73 |
> > >
> > > > Q1 How sensitive is performance to the number of domain experts and embedding dimensionality? Any heuristics or validation procedure beyond source-domain accuracy?
> > >
> > > In practice, we the number of experts are equal to source domain for we thinking it would engogh. The embedding are low than 32 dimensions.  To further discover this problems, we've  conducted more experiments and added discussions. The Follwings are the results and discussions.
> > >
> > >
> > > *Table 7. Sensitivity with fixed experts E=3 and varying embedding dimension d. Column-wise best in **bold**. Values are accuracies (%).*
> > >
> > > |  d   |     P     |     A     |     C     |     S     |  average  |
> > > | :--: | :-------: | :-------: | :-------: | :-------: | :-------: |
> > > |  5   |   95.81   |   84.03   |   78.71   |   70.73   |   82.32   |
> > > |  10  |   96.29   |   83.69   | **80.29** | **76.05** | **84.08** |
> > > |  15  | **96.53** | **84.08** |   69.24   |   71.54   |   80.35   |
> > > |  20  |   96.23   |   83.30   |   75.85   |   76.00   |   82.85   |
> > >
> > > *Table 8. Sensitivity with fixed embedding dimension d=10 and varying number of experts E. Column-wise best in **bold**. Values are accuracies (%).*
> > >
> > > |  E   |     P     |     A     |     C     |     S     |  average  |
> > > | :--: | :-------: | :-------: | :-------: | :-------: | :-------: |
> > > |  1   |   95.51   |   83.11   |   75.00   |   74.65   |   82.07   |
> > > |  3   |   96.29   |   83.69   | **80.29** | **76.05** | **84.08** |
> > > |  4   |   95.27   |   83.59   |   77.52   |   68.92   |   81.33   |
> > > |  5   |   95.51   |   84.28   |   75.13   |   73.15   |   82.02   |
> > > |  6   |   96.23   | **85.06** |   73.98   |   73.91   |   82.29   |
> > > |  7   |   95.45   |   84.08   |   74.62   |   73.63   |   81.94   |
> > > |  10  | **96.35** |   83.98   |   78.92   |   73.63   |   83.22   |
> > >
> > > Sensitivity is non–monotonic in both factors. With experts fixed at E=3 (Table A), a medium environment embedding performs best: d=10 attains the highest average 84.08%, whereas a smaller embedding underfits (d=5: 82.32%). With embedding fixed at d=10 (Table B), increasing experts from E=1 to E=3 improves the average from 82.07% to 84.08%; further increases provide no consistent gains (e.g., E=10: 83.22%). A few well–specialized experts (E≈3) together with a compact environment embedding (d≈10) strike a reliable balance between expressivity and out–of–domain generalization. To summarize: BAG is not sensitive to these two parameters, and the performance is within a controllable range; even suboptimal parameter choices do not push it below baselines such as ERM, IRM and ACTIR.
> > >
> > >
> > >
> > > The number and embedding dimensions can be validated by train/validation split. Detaily, some part like 10% souce domain data can be used as validation datasets for discover correct paraments.

---

> > > > ### Author Response · Authors · 2025-11-22
> > > > **Reply to question 2**
> > > >
> > > > > Q2 If p(e∣b) is wrong (e.g., target domain outside the convex hull of sources), how does BAG degrade? Any fallback to invariant-only predictions?
> > > >
> > > > You question are siginacantly important. In real world task, As showed in the ablation study, without the environment routing mechanism, our method is still good enough. Since it's hard to test this fallback in real word datasets, we conducted an experiment on a synthetic dataset to assess how a wrong $P(e \mid b)$ affects our model.
> > > > Specificially We modified the original data generation pipeline as follows. When constructing the bias variable $b$, instead of using a linear combination of the stable content $C$ and the embedding of the environment variable $E$, we replaced $E$ with Gaussian noise $\epsilon \sim \mathcal{N}(0, \sigma^{2} I)$. This change greatly reduces the amount of information about $e$ that is directly embedded in $b$.
> > > >
> > > > As shown in the table below, when the information content in $P(e \mid b)$ is reduced, the performance of BAG degrades, which is consistent with the intended stress test. Even so, BAG remains stronger than standard ERM under this setting.
> > > >
> > > >
> > > > | Ablation | BAG baseline | Wrong P(e\|b) |
> > > > | -------- | ------------ | ------------- |
> > > > | Acc      | 97.48        | 63.00         |

---

### Official Review · Reviewer_1V8g · 2025-11-02

**Soundness:** 2
**Presentation:** 2
**Contribution:** 2
**Rating:** 4
**Confidence:** 4

**Summary:**

This paper challenges the common assumption in OOD generalization that spurious or bias features should always be removed. The authors provide theoretical conditions under which bias can be identifiable and beneficial for prediction, particularly when there exists an unblocked causal path between bias and label. Building on this, they propose Bias-Aware Generalization (BAG) — an end-to-end framework that disentangles invariant and bias representations via a VAE with an independence regularizer, routes predictions through inferred environments, and corrects bias using an adaptive, learnable label prior. Experiments on synthetic and standard DG benchmarks (PACS, OfficeHome) show modest gains over ERM.

**Strengths:**

* **Interesting and original perspective** — The paper takes a refreshing stance by arguing that bias or spurious features are not always detrimental. This idea of *leveraging* bias for better OOD generalization is both conceptually interesting and relevant, especially given how dominant the “bias elimination” mindset has been in this field.

* **Solid theoretical framing** — The authors provide a clear theoretical justification for when and why bias can be beneficial, using identifiability conditions and causal reasoning. This helps ground the idea beyond intuition and makes the argument more rigorous than many prior discussions on bias utilization.

* **Coherent and well-structured framework** — The proposed BAG model integrates disentanglement, environment routing, and adaptive label priors into a unified end-to-end architecture. The framework is logically consistent and easy to follow, with each component corresponding naturally to a part of the theoretical motivation.

**Weaknesses:**

* **Lack of ablations and component analysis** — The paper does not provide sufficient ablation studies to isolate where the performance gain comes from. For instance, it’s unclear how much each part—the content predictor, bias predictor, environment routing module, or adaptive prior—actually contributes. The absence of results like *C-only vs. B-only*, or *with vs. without routing and prior correction*, makes it hard to judge whether the added modules are necessary or if the model mainly reproduces SFB’s effect in a different architecture.

* **Limited experimental scope and dataset diversity** — The experiments are conducted only on synthetic data, PACS, and OfficeHome. These are relatively small and saturated benchmarks that no longer reflect the challenges of modern OOD evaluation. The paper should consider following the **DomainBed** codebase and benchmark protocol (VLCS, PACS, OfficeHome, TerraIncognita, DomainNet) to ensure fair hyperparameter tuning and reproducible comparisons. In addition, it would be helpful to include datasets that explicitly test spurious correlations—such as **CMNIST**, **Waterbirds**, **CelebA**, **MetaShift**, or **BAR** (as used in *Project-Probe-Aggregate: Efficient Fine-Tuning for Group Robustness*)—to more convincingly demonstrate the benefit of leveraging bias. Currently, the performance improvements appear substantial on synthetic data but modest on real-world settings.

* **Overlap with prior work and unclear novelty** — The overall framework, especially the test-time refinement step and bias correction mechanism, is conceptually very close to Stable Feature Boosting (SFB). While the paper adds disentanglement and environment routing, it is not yet clear whether these bring substantial new insights or measurable benefits. Without stronger theoretical or empirical differentiation, the work feels more like an incremental extension rather than a clear step forward.

**Questions:**

1. Could the authors provide detailed abations to show the independent effect of the content predictor, bias predictor, environment routing module, and adaptive label prior? Which component contributes most to the final improvement?

2. Is the environment predictor $p(e|b)$ truly necessary? Would directly predicting $p(y|b)$ yield similar performance? How sensitive is the model to the number of experts $M$ in the routing mechanism?

3. How does the proposed predictor refinement differ from SFB’s bias correction step, both theoretically and empirically? Could the authors clarify the key advantage that BAG offers over SFB beyond architectural integration?

4. Are the experiments implemented within the DomainBed codebase for fair comparison? Do the authors plan to include larger or spurious-correlation datasets (e.g., Waterbirds, CelebA, CMNIST) to validate the claim that bias utilization helps in realistic settings?

---

> ### Author Response · Authors · 2025-11-21
> **Overall reply and replies to question 1 and Weekness 1**
>
> Dear Reviewer 1V8g,
>
> We sincerely thank you for your thorough review and thoughtful feedback. We are grateful for your recognition of our perspective, theoretical analysis, and the BAG framework. We have added a lot of additional experiments. Please see the point by point responses below.
>
> > Q1.Could the authors provide detailed abations to show the independent effect of the content predictor, bias predictor, environment routing module, and adaptive label prior? Which component contributes most to the final improvement?
> >
> > W1.**Lack of ablations and component analysis** — The paper does not provide sufficient ablation studies to isolate where the performance gain comes from. For instance, it’s unclear how much each part—the content predictor, bias predictor, environment routing module, or adaptive prior—actually contributes. The absence of results like *C-only vs. B-only*, or *with vs. without routing and prior correction*, makes it hard to judge whether the added modules are necessary or if the model mainly reproduces SFB’s effect in a different architecture.
>
> Thanks for suggesting the detailed discussion of effect of different parts of BAG. In the original submission, we have ablation studies on PACS showing that the VAE, bias-aware prediction with the environment routing mechanism, and bias correction with an adaptive prior all contribute to performance. From Table 5 in Appendix E.2, environment routing mechanism contributes the most among environment routing mechanism and bias correction with an adaptive prior. Following your suggestions, we have added more results about the performance of content predictor and bias predictor in Table 6. Also we added discussions about performance of each part in Appendix E.2. The related discussion and table in the revised version are as follows.
>
> Table 6. Different parts performance results on PACS. Values are accuracies (%).
>
> | Method                 |     P |     A |     C |     S |   Avg |
> | ---------------------- | ----: | ----: | ----: | ----: | ----: |
> | Content predictor Only | 95.57 | 81.10 | 75.00 | 67.83 | 79.88 |
> | Bias predictor Only    | 94.01 | 73.58 | 78.67 | 66.23 | 78.12 |
> | BAG                    | 96.29 | 83.69 | 80.29 | 76.05 | 84.08 |
>
> Table 5 presents a systematic ablation of BAG on PACS and in our synthetic simulations. The full method comprises two optimization stages and a VAE module. In the ablations, we remove exactly one component at a time, as indicated by a hyphen in the variant name: **BAG-VAE** drops the **VAE** in favor of a linear projection; **BAG-TTA** removes the **T**est **T**ime **A**daptation that uses pseudo-labels to guide the bias branch; and **BAG-RE** removes the **R**eweight **E**xperts mechanism. For transparency on the predictive roles of the two branches,Table 6 further reports the behavior of the bias predictor and the content predictor when used in isolation.
>
> Empirically, all BAG variants outperform ERM across datasets, confirming the effectiveness of our bias utilization design. **BAG-RE** and **BAG-TTA** show complementary strengths and weaknesses, consistent with data–specific factors such as domain composition and the severity of distribution shift, whereas **BAG-VAE** lags behind the full BAG, underscoring the value of accurately identifying and disentangling $\mathbf{b}$ (bias aligned factors) and $\mathbf{c}$(content aligned factors). Using only the bias predictor or only the content predictor yields average accuracies of roughly $78\%$ and $80\%$, respectively; each alone lacks sufficient information and, without a learnable class prior, is vulnerable to label shift. Once the two predictors are combined under a learnable label prior, performance rises to state of the art levels among the baselines considered.
>
> In summary, the ablations show that each component contributes meaningfully: (i) explicit disentanglement of bias and content, (ii) expert reweighting and test–time adaptation to stabilize routing, and (iii) a compact generative prior via the VAE. Together, these elements enable BAG to deliver robust and consistently strong domain generalization performance, while simplified variants remain competitive and illuminate the role of each design choice.

---

> > ### Author Response · Authors · 2025-11-21
> > **Reply to question 2**
> >
> > > Q2. Is the environment predictor truly necessary? Would directly predicting yield similar performance? How sensitive is the model to the number of experts in the routing mechanism?
> >
> > Thank you for focusing on the environment predictor. The environment predictor is necessary, this is because directly using bias to predict $b$ to predict $y$ can be dangerous because $p(y|b)$ changes due to label shift. Directly predicting may cause problems which is somehow a widely point so many works just seeking invariant learning. In practice, the number of experts are equal to source domain number for we thinking it would enough. To further discuss this question, we conducted some experiments like follows and have added discussion like follows in Appendix E.3.
> >
> >
> > *Table 7. Sensitivity with fixed experts E=3 and varying embedding dimension d. Column-wise best in **bold**. Values are accuracies (%).*
> >
> > |  D   |     P     |     A     |     C     |     S     |  average  |
> > | :--: | :-------: | :-------: | :-------: | :-------: | :-------: |
> > |  5   |   95.81   |   84.03   |   78.71   |   70.73   |   82.32   |
> > |  10  |   96.29   |   83.69   | **80.29** | **76.05** | **84.08** |
> > |  15  | **96.53** | **84.08** |   69.24   |   71.54   |   80.35   |
> > |  20  |   96.23   |   83.30   |   75.85   |   76.00   |   82.85   |
> >
> > *Table 8. Sensitivity with fixed embedding dimension d=10 and varying number of experts E. Column-wise best in **bold**. Values are accuracies (%).*
> >
> > |  E   |     P     |     A     |     C     |     S     |  average  |
> > | :--: | :-------: | :-------: | :-------: | :-------: | :-------: |
> > |  1   |   95.51   |   83.11   |   75.00   |   74.65   |   82.07   |
> > |  3   |   96.29   |   83.69   | **80.29** | **76.05** | **84.08** |
> > |  4   |   95.27   |   83.59   |   77.52   |   68.92   |   81.33   |
> > |  5   |   95.51   |   84.28   |   75.13   |   73.15   |   82.02   |
> > |  6   |   96.23   | **85.06** |   73.98   |   73.91   |   82.29   |
> > |  7   |   95.45   |   84.08   |   74.62   |   73.63   |   81.94   |
> > |  10  | **96.35** |   83.98   |   78.92   |   73.63   |   83.22   |
> >
> > The environmental predictor is necessary. Directly predicting \(y\) from bias–aligned features \(b\) effectively fits \(p(y | b)\), which is brittle under label shift: when \(p(y)\) changes across domains, the conditional \(p(y | b)\) drifts, and target performance degrades. Our approach explicitly estimates a latent environment and routes each example to domain conditioned experts, separating environment specific variation from task–relevant signals and improving robustness. As shown in the ablation table (method BAG–RE), removing the environmental predictor reduces the mean accuracy to 81.8%, below that of the full model. In our hyperparameter sweeps (Tables A and B), most configurations that include the predictor exceed 81.8% on average, and the best setting with E=3, d=10 reaches 84.08% with strong performance on the challenging domains C=80.29% and S=76.05%. In summary, conditioning the classifier on an explicitly predicted environment addresses distribution shift and yields measurable, domain–wise robust gains over direct prediction.
> >
> > Sensitivity is non monotonic in both factors. With experts fixed at E=3 (Table A), a medium environment embedding performs best: d=10 attains the highest average 84.08%, whereas a smaller embedding underfits (d=5: 82.32%). With embedding fixed at d=10 (Table B), increasing experts from E=1 to E=3 improves the average from 82.07% to 84.08%; further increases provide no consistent gains (e.g., E=10: 83.22%). A few well–specialized experts (E≈3) together with a compact environment embedding (d≈10) strike a reliable balance between expressivity and out–of–domain generalization. To summarize: BAG is not sensitive to these two parameters, and the performance is within a controllable range; even suboptimal parameter choices do not push it below baselines such as ERM, IRM and ACTIR.

---

> ### Author Response · Authors · 2025-11-21
> **Replies to question 3, 4 and weakness 2,3**
>
> > Q3. How does the proposed predictor refinement differ from SFB’s bias correction step, both theoretically and empirically? Could the authors clarify the key advantage that BAG offers over SFB beyond architectural integration?
> >
> > W3. Overlap with prior work and unclear novelty — The overall framework, especially the test-time refinement step and bias correction mechanism, is conceptually very close to Stable Feature Boosting (SFB). While the paper adds disentanglement and environment routing, it is not yet clear whether these bring substantial new insights or measurable benefits. Without stronger theoretical or empirical differentiation, the work feels more like an incremental extension rather than a clear step forward.
>
> Thanks for the concern about the difference. Here we would like to highlight our difference and advantages. In theory, SFB did not come to the latent-variable level; it splits the representation into invariant and biased components, then shows that the marginal probabilities can be decomposed and that test-time adaptation of the biased component can work. We came into latent variables level and provide identifiability, also we consider environment routing mechanism and label shift. We show two ways to better use bias even under label shift and the invariant may not too accurate.
>
> Empirically, the SFB method divide learned features into invariant and variant, and using the predictions based on the invariant features as pseudo-labels for test-time adaptation of the biased component. We use a generative model that factorizes representations into invariant content and contextual bias, and later use the bias features to estimate environment states and design an environment routing mechanism that leverages these states to gate a mixture of predictors, each conditioned on a specific environment. We extend the bias correction in SFB with an adaptive label prior to solve label shift problem.
>
> Here we would like to highlight three key advantages: (1) We use a generative model to better learn latent variables with identifiability. (2) We consider label shift, which is a more general case, and use an adaptive label prior to solve it. (3) We provide an environment routing mechanism to better use bias without relying solely on the invariant predictor.
>
> > Q4. Are the experiments implemented within the DomainBed codebase for fair comparison? Do the authors plan to include larger or spurious-correlation datasets (e.g., Waterbirds, CelebA, CMNIST) to validate the claim that bias utilization helps in realistic settings?
> >
> > W2. **Limited experimental scope and dataset diversity** — The experiments are conducted only on synthetic data, PACS, and OfficeHome. These are relatively small and saturated benchmarks that no longer reflect the challenges of modern OOD evaluation. The paper should consider following the **DomainBed** codebase and benchmark protocol (VLCS, PACS, OfficeHome, TerraIncognita, DomainNet) to ensure fair hyperparameter tuning and reproducible comparisons. In addition, it would be helpful to include datasets that explicitly test spurious correlations—such as **CMNIST**, **Waterbirds**, **CelebA**, **MetaShift**, or **BAR** (as used in *Project-Probe-Aggregate: Efficient Fine-Tuning for Group Robustness*)—to more convincingly demonstrate the benefit of leveraging bias. Currently, the performance improvements appear substantial on synthetic data but modest on real-world settings.
>
> Thank you for suggesting more experiments. We don't use DomainBed codebase, this is because we follow setting of SFB which is the most related work as we know. So the comparison is fair. Our code base is from ACTIR as we can't found code of SFB but SFB said the their codebase is from ACTIR .
>
> In light of your suggestions, we have added experiments on DomainNet[1], which is a large-scale dataset. We compared BAG against baselines like ERM, MMD[5], IRM[7], ACTIR[10], GMDG[11], MIRO[2], RDM[3], DANN[4], CDANN[6], SagNet[8], EQRM[9]. You can find them in Table 3 in Section 5.2 in the revised version.
>
> We are currently doing experiments on the Waterbirds dataset. Because our focus is domain generalization and Waterbirds is primarily a spurious-correlation benchmark, we need more time to adapt our code (e.g., the dataloader and related components). We will treat this as our top priority and will share the results as soon as the experiments are completed.

---

> > ### Author Response · Authors · 2025-11-21
> > **Results and discussion on DomainNet**
> >
> > **DomainNet target-domain accuracies (%) — ResNet-50 backbone** *Most baseline results are taken from [9].*
> >
> > | Algorithm | Year |  clipart | infograph | painting | quickdraw |     real |   sketch |      Avg |
> > | --------- | :--: | -------: | --------: | -------: | --------: | -------: | -------: | -------: |
> > | MIRO[2]   | 2022 |       -- |        -- |       -- |        -- |       -- |       -- |     44.3 |
> > | RDM[3]    | 2024 |       -- |        -- |       -- |        -- |       -- |       -- |     43.4 |
> > | ERM       |  —   |     58.1 |      18.8 |     46.7 |      12.2 |     59.6 |     49.8 |     40.9 |
> > | DANN[4]   | 2016 |     53.1 |      18.3 |     44.2 |      11.8 |     55.5 |     46.8 |     38.3 |
> > | MMD[5]    | 2018 |     32.1 |      11.0 |     26.8 |       8.7 |     32.7 |     28.9 |     23.4 |
> > | CDANN[6]  | 2018 |     54.6 |      17.3 |     43.7 |      12.1 |     56.2 |     45.9 |     38.3 |
> > | IRM[7]    | 2019 |     48.5 |      15.0 |     38.3 |      10.9 |     48.2 |     42.3 |     33.9 |
> > | SagNet[8] | 2021 |     57.7 |      19.0 |     45.3 |      12.7 |     58.1 |     48.8 |     40.3 |
> > | EQRM[9]   | 2022 |     56.1 |      19.6 |     46.3 |      12.9 |     61.1 |     50.3 |     41.0 |
> > | ACTIR[10] | 2022 |     50.0 |      22.6 |     43.5 |      11.7 |     57.8 |     46.8 |     38.7 |
> > | GMDG[11]  | 2024 | **63.4** |      22.4 |     51.4 |      13.4 |     64.4 | **52.4** |     44.6 |
> > | **BAG**   | 2025 |     61.8 |  **25.6** | **51.8** |  **13.9** | **65.5** |     50.3 | **44.8** |
> >
> > We use a ResNet-50 backbone for feature extraction to match the baselines that also adopt ResNet-50. For MIRO [2] and RDM [3], we report only the mean accuracy because their papers provide averages without per-domain breakdowns. The ACTIR [10] result is obtained by running the authors’ public code under their official setting, as ACTIR does not report DomainNet results. On DomainNet, our method attains the best average accuracy (44.8%), surpassing recent works like GMDG [11] (44.6%) and MIRO [2] (44.3%). At the per domain level, it is best on infograph, painting, quickdraw, and real, while remaining competitive on clipart and sketch. The largest gain occurs on infograph, likely because background/stylistic artifacts are prominent there; explicitly disentangling content from bias both mitigates spurious domain cues and enables us to leverage bias when it is informative. On clipart and sketch we are not SOTA, which is likely because the domain-specific bias signal in these two domains is weak, uninformative, or misaligned, limiting the benefit of leveraging it. Compared with ACTIR [10] under the same setting, our method achieves substantially higher averages, which we attribute to more accurate disentanglement of, and principled use of, bias.
> >
> >
> >
> >
> >
> > [1] Peng X, Bai Q, Xia X, et al. Moment matching for multi-source domain adaptation[C]//Proceedings of the IEEE/CVF International Conference on Computer Vision. 2019: 1406–1415.
> >
> > [2] Cha J, Lee K, Park S, et al. Domain generalization by mutual-information regularization with pre-trained models[C]//European Conference on Computer Vision. Cham: Springer Nature Switzerland, 2022: 440–457.
> >
> > [3] Nguyen T, Do K, Duong B, et al. Domain generalisation via risk distribution matching[C]//Proceedings of the IEEE/CVF Winter Conference on Applications of Computer Vision. 2024: 2790–2799.
> >
> > [4] Ganin Y, Ustinova E, Ajakan H, et al. Domain-adversarial training of neural networks[J]. Journal of Machine Learning Research, 2016, 17(59): 1–35.
> >
> > [5] Li H, Pan S J, Wang S, et al. Domain generalization with adversarial feature learning[C]//Proceedings of the IEEE Conference on Computer Vision and Pattern Recognition. 2018: 5400–5409.
> >
> > [6] Long M, Cao Z, Wang J, et al. Conditional adversarial domain adaptation[J]. Advances in Neural Information Processing Systems, 2018, 31.
> >
> > [7] Arjovsky M, Bottou L, Gulrajani I, et al. Invariant risk minimization[J]. arXiv preprint arXiv:1907.02893, 2019.
> >
> > [8] Nam H, Lee H J, Park J, et al. Reducing domain gap by reducing style bias[C]//Proceedings of the IEEE/CVF Conference on Computer Vision and Pattern Recognition. 2021: 8690–8699.
> >
> > [9] Eastwood C, Robey A, Singh S, et al. Probable domain generalization via quantile risk minimization[J]. Advances in Neural Information Processing Systems, 2022, 35: 17340–17358.
> >
> > [10] Jiang Y, Veitch V. Invariant and transportable representations for anti-causal domain shifts[J]. Advances in Neural Information Processing Systems, 2022, 35: 20782–20794.
> >
> > [11] Tan Z, Yang X, Huang K. Rethinking multi-domain generalization with a general learning objective[C]//Proceedings of the IEEE/CVF Conference on Computer Vision and Pattern Recognition. 2024: 23512–23522.

---

### Official Review · Reviewer_Ns8F · 2025-11-03

**Soundness:** 3
**Presentation:** 1
**Contribution:** 1
**Rating:** 0
**Confidence:** 5

**Summary:**

This paper challenges the typical view that bias should be minimized for robust out-of-distribution (OOD) generalization. The authors propose Bias-Aware Generalization (BAG), a framework that leverages both stable and bias features to improve prediction performance. Through experiments on synthetic and real-world datasets, BAG outperforms traditional methods like ERM and IRM. The approach combines stable feature predictors with bias terms and uses a mixture of experts model, enhanced by soft labeling for test-time adaptation, showing that bias, when carefully managed, can enhance OOD generalization.

**Strengths:**

(1) The fundamental idea is reasonable. Covariate bias is indeed influential in classification and OOD-related tasks. I believe the authors’ perspective is justified. However, I believe this can be both helpful and risky. it may improve performance on certain tasks but could also lead to biased errors in other scenarios.


(2) The authors conduct experiments on both synthetic and real-world datasets, demonstrating the effectiveness of their proposed method compared to existing approaches.

**Weaknesses:**

I have reviewed this paper for NeurIPS 2025, in which 5 reviewers unanimously decided to reject the paper.

In my previous review, I outlined several issues, but I found that the authors did not make substantial revisions. Even minor issues, such as typographical errors in symbols and writing, were not corrected.

Given these considerations, I have decided to reject the paper. In today’s world, where the volume of peer review work is high, I strongly recommend that the authors take each reviewer’s feedback seriously and make substantial improvements addressing the weaknesses of the work before resubmitting. The fllowing reviews also included some previous review comments that I agree with, for reference.

(1) The decomposition of content and bias representations is not empirically validated. It remains a verbal argument. How can one verify that the learned representations actually correspond to the intended content and bias factors? This is a key issue.

(2) A central part of the method’s separation of bias and content relies on the independence loss. However, this only enforces a necessary, not sufficient, condition for conditional independence.

(3) The core of the paper lies in validating the benefit of explicitly modeling and utilizing bias, which seems to be evaluable without using test-time adaptation setting. I do not see the necessity of including test-time adaptation component. If I understand correctly, the proposed method is the only one that uses test time adaptation, which gives it an unfair advantage in the empirical evaluation.

(4) I recommend that the authors use the same set of baselines across the benchmarks on which they conduct experiments. Although the authors mention that SFB did not report performance on the Office-Home dataset, it still seems feasible to compare the performance of baselines like GMDG on the PACS dataset.

(5) Large part of the claimed contributions of the paper have been derived in Stable Feature Boosting (SFB) already.

(6) Similarly, a mixture of experts method that is not too dissimilar to what the authors propose has been proposed in Prashant et al. ‘Scalable out-of-distribution robustness in the presence of unobserved confounders’ (2025), but is not referenced and at least conceptually compared to.

(7) In line 1200 of the pseudocode, is there a typo? Should it be f_c(\mathbf C) instead of f_c \mathbf C?

(8) I suggest the authors visualize e b y c x in the clearest way possible. I cannot distinguish the difference between e and b. You assume b ⊥ c | y (line 207), and also assume y is not independent of e given c (lines 245-246), but I fail to understand the rationale behind these assumptions.

(9) The writing is unclear, and there are some necessary explanations for certain concepts that would be meaningful. For example, in Lemma 2.1, what does positive support refer to, and why is this emphasized? Isn't positive support self-evident? What does $\mathcal{B}$ mean in the context of domain variability? This is not explained. Additionally, what does linear independence mean, and why is it reasonable?

(10) it is unclear what "RE" in BAG-RE refers to (w/o Regularization of IND?), and what BAG-VAE stands for. Isn't your BAG method based on VAE already? what is BAG without VAE refers to? None of these are clearly explained, and the writing is very unclear.

**Questions:**

see above

---

> ### Author Response · Authors · 2025-11-21
> **Overall reply and replies to weakness (1) and (2)**
>
> Dear Reviewer Ns8F,
>
> Thank you for your acknowledgment of our perspective and experimental performance, and thank you for raising these questions to help improve our paper. We would like to respectfully emphasize that we greatly appreciate all the reviewers’ comments and have made every effort to revise the paper accordingly. While we were unable to follow all of the reviewers’ comments, as some seem to stem from misunderstandings, we have carefully revised the manuscript to clarify our statements (to avoid the misunderstanding) and have incorporated constructive and reasonable suggestions.
> For example, in this submission, we **have rewritten** the paper with more discussion of SFB in **Section 1** to compare with the most related work and state contributions, **we newly added Section 3** to fully introduce preliminary work, and reorganized **Sections 2 and 4** to highlight when and how we can use bias. Specifically, **weaknesses (2), (3), (5), and (10) are addressed**, which can be found in **lines 262, 270 (2); lines 90, 393 (3); lines 70, 80, 317, Section 3 (5); Appendix E.2 (10)**. For (4) and (6), the recommended papers do not fit well for a fair comparison (e.g., GMDG uses a different network architecture than our method and SFB). For (7), we missed this comment in the modification and highly appreciate you pointing this typo out. We have tried our best to work together with all reviewers to make the new submission better.
>
> Below, we provide detailed responses to each question. We sincerely hope that our efforts can be noticed and the explanations can adequately address your concerns and questions.
>
> >  (1) The decomposition of content and bias representations is not empirically validated. It remains a verbal argument. How can one verify that the learned representations actually correspond to the intended content and bias factors? This is a key issue.
>
> Thanks for raising this concern. In light of your suggestion, we have added a visualization of the learned representations to show how the content and bias are decomposed. Please refer to **Figure 4 (Appendix E.4)** for details.
>
> Specifically, we qualitatively assess disentanglement on PACS by visualizing Grad-CAM[1] attributions for the content branch ($\mathbf{c}$) and the bias branch ($\mathbf{b}$). Using the trained checkpoint and backbone, we wrap the BAG model to expose logits $f_\beta$ (for $\mathbf{c}$) and $f_\eta$ (for $\mathbf{b}$), target the last `layer4` block, and compute two maps with `ClassifierOutputTarget` set to the ground-truth class. Each map is resized to the image resolution, min–max normalized to $[0,1]$, and overlaid (red for $\mathbf{c}$, blue for $\mathbf{b}$); panels are shown as: original, $\mathbf{b}$, $\mathbf{c}$, composite. Visually, $\mathbf{c}$ concentrates on object regions while $\mathbf{b}$ emphasizes background/style cues, confirming the intended decomposition.
>
> We did not include these results in our initial submission because the decomposition of content and bias is not our main contribution [1]. To avoid overclaiming, we did not present these results and instead cited the decomposition as a lemma rather than a theorem. In the revised version, we have added these results to the appendix and explicitly clarified that the decomposition is not our contribution, to prevent any potential misunderstanding.
>
> > (2) A central part of the method’s separation of bias and content relies on the independence loss. However, this only enforces a necessary, not sufficient, condition for conditional independence.
>
> Thanks for pointing out this. We would like to highlight that the separation of bias and content **DO NOT** only rely on the independence loss. As shown in **line 270**, the independence loss is a loss to encourage the independence but cannot provide guarantee. Instead, we use a generative model with the identification guarantee to achieve the distentanglement, as mentioned in **line 262**. Intuitively, such distentanglement is achieved by constraining both the reconstruction (no infomation loss), and conditional independence with the KL divergence (aligning the posterior distribution of the learned variables with a conditionally independent prior). We recommend the reader to Kong et al. [2] for more details about such implenmentation, and be more than happy to discuss more if there is anything unclear.
>
> [1]Selvaraju R R, Cogswell M, Das A, et al. Grad-cam: Visual explanations from deep networks via gradient-based localization[C]//Proceedings of the IEEE international conference on computer vision. 2017: 618-626.
>
> [2]Kong L, Xie S, Yao W, et al. Partial identifiability for domain adaptation[J]. arXiv preprint arXiv:2306.06510, 2023.

---

> ### Author Response · Authors · 2025-11-21
> **Replies to weakness (3) -(6)**
>
> > (3) The core of the paper lies in validating the benefit of explicitly modeling and utilizing bias, which seems to be evaluable without using test-time adaptation setting. I do not see the necessity of including test-time adaptation component. If I understand correctly, the proposed method is the only one that uses test time adaptation, which gives it an unfair advantage in the empirical evaluation.
>
>
> Thank you for this concern. The test-time adaptation is part of two ways to use bias better, so it is needed to use pseudo labels to guide bias usage. The test-time adaptation is related to **Section 4.3** and described in detail. Also, in **line 90** we mentioned the effects of this. The most related work, BAG, uses test-time adaptation. As we followed the settings in BAG mentioned in **line 393**, test-time adaptation here is totally fair.
>
> > (4) I recommend that the authors use the same set of baselines across the benchmarks on which they conduct experiments. Although the authors mention that SFB did not report performance on the Office-Home dataset, it still seems feasible to compare the performance of baselines like GMDG on the PACS dataset.
>
> Thanks for this recommendation. We used the settings following SFB and ACTIR as we mentioned in **line 393**. We contacted the SFB team but didn’t get the code, so we lack the results on the Office-Home dataset. We used code and settings originally from ACTIR. We didn’t find GMDG results with a ResNet-18 backbone which most baselines used as mentioned in **line 446** on the PACS dataset. So we don’t add results about it for a fair comparison. To further solve this question, in the revised version, we used the **code of GMDG to do experiments with ResNet-18** and have added results in Table 2. We also added more experiments on a larger-scale dataset, DomainNet, in which we added more baselines to indicate the performance of our method. The result can be found in Table 4. For fair comparation, we have **used the code of ACITR to do experiments in the Office-Home dataset and Domainnet** and include the result in Table 1 and 3. It can be found that BAG outperform this baseline of same settting in all datasets.
>
>
> > (5) Large part of the claimed contributions of the paper have been derived in Stable Feature Boosting (SFB) already.
>
> Thank you for emphasizing this issue again. This may have been raised before, but we have rewritten the paper to show the comparison and restate our contributions. In **paragraphs starting from line 70**, we have discussed SFB and pointed out some limitations. In **paragraphs starting from line 80**, we highlight differences between the two methods in implementation. We also **specifically** added **Section 3** to show the preliminary work in SFB, and all related SFB contributions are presented in this section to avoid misunderstanding. We also compared SFB and presented our contributions in **Section 4.3**, as in **paragraphs starting from line 317**, and in the newly proposed **Theorem 4.2**. Here we would like to **briefly** highlight our key contributions compared with SFB. We consider label shift, which is a more general setting, so we use test-time adaptation with a learnable label prior, and a theoretical upper bound is given. We adopt a generative model that factorizes representations into invariant content and contextual bias in an identifiable manner. Also, we add the environment routing mechanism to better use bias if the invariant content may be under-learned.
>
> > (6) Similarly, a mixture of experts method that is not too dissimilar to what the authors propose has been proposed in Prashant et al. ‘Scalable out-of-distribution robustness in the presence of unobserved confounders’ (2025), but is not referenced and at least conceptually compared to.
>
> Thanks for pointing out this paper. This paper is centrally focused on identifiability theory and has no results on domain generalization dataset like PACS or Office-Home. Since we don’t claim contribution in identifiability and our method is not limited to a mixture of experts, we don’t compare it to show our main contribution. If it’s necessary from your insights, we will add it in the related works.

---

> ### Author Response · Authors · 2025-11-21
> **Replies to weakness (7) -(10)**
>
> > (7)In line 1200 of the pseudocode, is there a typo? Should it be f_c(\mathbf C) instead of f_c \mathbf C?
>
> Thanks for your advice. It's should be $f_c(\mathbf C)$.  We missed this point partly since due to page limit we have to put the core pseudocode in appendix. We sincerely thank you again for your careful check. We promise we will learn a lesson from it and avoid similar problems.
>
> > (8) I suggest the authors visualize e b y c x in the clearest way possible. I cannot distinguish the difference between e and b. You assume b ⊥ c | y (line 207), and also assume y is not independent of e given c (lines 245-246), but I fail to understand the rationale behind these assumptions.
>
> Thanks.  Here we would like to explain these concepts and relations in detail. We’ve visualized all variables and relations in Figure 2. Also, the detailed meanings of the variables can be found in **Section 2.1**. The $\mathbf{b}$ means **b**ias, $\mathbf{c}$ means **c**ontent, $e$ means **e**nvironment, $\mathbf{x}$ is the observation, and $y$ represents the labels. $\mathbf{b} \perp \mathbf{c} \mid y$ and $y \not\!\perp e \mid \mathbf{c}$ can both be found in Figure 2  intuitively. Here we would like to share motivation as follows: The content (invariant) is just related to $y$, so it’s only decided by $y$, which suggests that $\mathbf{b} \perp \mathbf{c} \mid y$. For example, if the label is animal species, the content (what an animal looks like) is only decided by its species and is independent of the environment if the species is fixed (like a row). There is label shift, which indicates that $y$ is influenced by environments $e$; for example, species distributions differ across environments. $y$ will not be independent of $e$ even given $\mathbf{c}$, since there is a direct path from $e$ to $y$, so we can use $e$ to better predict $y$.
>
> > (9) The writing is unclear, and there are some necessary explanations for certain concepts that would be meaningful. For example, in Lemma 2.1, what does positive support refer to, and why is this emphasized? Isn't positive support self-evident? What does B mean in the context of domain variability? This is not explained. Additionally, what does linear independence mean, and why is it reasonable?
>
> Thank you for your reading for our theory. Here, since we don’t claim contributions on identifiability, we just use the lemma to show key ideas. We didn’t say a lot about Lemma 2.1 because it’s not our contribution and pages are limited. We’ve added it in the Appendix B.2, and, if possible, we will add some expansion in the main text if reviewers think it’s needed and page space is enough. Here we show the newly added explanations.
>
> We would like to detailedly explain some concepts in Lemma 2.1 as follows:  (1)Here “Positive support” means the set of all points where a probability distribution assigns strictly positive probability or density (i.e., $p(x) > 0$). This is emphasized because the probability distribution may have some point in 0 for example the distribution are uniform distribution. To maintain academic rigor, this hypothesis was highlighted. (2)The $\mathcal{B}$ is introduced in A3 and it  refers to the set (or space) of domain-specific latent variables.   (3)Linear independence means that changes in different environments produce distinct, non-redundant effects on the domain-specific latent variable $b$; none of these variations can be expressed as a combination of others. This is a reasonable assumption because, with multiple diverse environments, it ensures that each domain provides unique information about how $b$ varies, making the latent factors identifiable rather than entangled.  The detailed explanations of this lemma can be found in paper[1]
>
> > (10)It is unclear what "RE" in BAG-RE refers to (w/o Regularization of IND?), and what BAG-VAE stands for. Isn't your BAG method based on VAE already? what is BAG without VAE refers to? None of these are clearly explained, and the writing is very unclear.
>
> Thank you for the attention to ablation study. In **Appendix E.2**, we’ve explained in detail that “-” means that, in the ablation study, we drop the corresponding part. For example, in BAG-VAE, we drop the VAE loss and just use a linear layer. RE means environment routing mechanism, and TTA means the test-time adaptation part. We **also** added some explanations about these in the main text in the newest version since we have one more page. We show the content as follows.
>
> We conduct an ablation study to isolate the effects of each optimization stage and the VAE in **BAG**. A leading “-” in a variant name means we remove only the corresponding component while keeping everything else unchanged. Specifically, **BAG-VAE** drops the **VAE** loss and uses a simple linear layer instead; **BAG-TTA** removes **T**est **T**ime **A**daptation and thus does not use pseudo labels to guide the bias branch; and **BAG-RE** removes the **R**eweight **E**xperts component of our model.

---

### Official Review · Reviewer_e4v7 · 2025-11-11

**Soundness:** 3
**Presentation:** 4
**Contribution:** 3
**Rating:** 6
**Confidence:** 2

**Summary:**

The paper challenges the conventional view in out-of-distribution (OOD) generalization that bias must always be removed. Instead, it argues that bias can sometimes help generalization and proposes a framework (BAG) that explicitly models and leverages bias.
The authors provide a theoretical analysis identifying conditions under which bias can aid prediction,  bias-aware generative model that disentangles invariant content and bias features, and a mixture-of-experts mechanism for bias-aware prediction, guided by an adaptive label prior to handle both covariate and label shift.

**Strengths:**

- The paper has a fairly decent novelty contribution, challenging entrenched assumptions in DG research about bias should be eliminated or not and reframes bias as a potentially beneficial signal. Furthermore, the formalization of when bias helps prediction (via “unblocked influence”) and the proofs of identifiability and performance are useful for technical depth and novelty. The use of causal graphs and conditional independencies is well-motivated.
- BAG unifies multiple strands: causal representation learning, mixture-of-experts modeling, and adaptive label calibration — into a coherent probabilistic framework.
- Experiments are well-structured and make sense for the pipeline.
- Paper is well written and structured.

**Weaknesses:**

- Only tests the problem on very small datasets, making it difficult to understand or interpret whether this framework is generalisable.
- The theoretical identifiability conditions (A1–A4) require smooth, positive densities and independent latent dimensions given (e, y). These are rarely satisfied in high-dimensional deep representations. A discussion of approximate or empirical identifiability would help.
- Although disentanglement is central, the paper lacks visualizations or examples showing what the learned bias and content dimensions represent in image space. It is a bit difficult to interpret some of the results due to this oversight.
- The gains seem to be mainly on the Synthetic case, where the non-synthetic examples have minimal gain that is often within the confidence threshold.

**Questions:**

A discussion of the theoretical assumptions (A1-A4) would be useful to understand where this is applicable, and where it is not (aka, which datasets/domains does this make sense to apply to, and which do not make sense?)

---

> ### Author Response · Authors · 2025-11-22
> **Overall reply and reply to weakness 1**
>
> Dear Reviewer e4v7,
>
> We sincerely appreciate your encouraging comments on our formulation, framework, and writing. Your valuable suggestions on presenting our performance and discussing our theory have significantly enhanced our paper. We have included the new results, pictures and discussions in the revised version. The one to one responses can be found below:
>
> > W1. Only tests the problem on very small datasets, making it difficult to understand or interpret whether this framework is generalisable.
>
>
> Thanks for your suggestion. We’ve added experiments on DomainNet[1], which is a large-scale dataset. We compared BAG against baselines ERM, MMD[5], IRM[7], ACTIR[10], GMDG[11], and added more baselines like MIRO[2], RDM[3], DANN[4], CDANN[6], SagNet[8], EQRM[9]. You can find them in Table 3 in Section 5.2 in the revised version.
>
> **DomainNet target-domain accuracies (%) — ResNet-50 backbone** *Most baseline results are taken from [9].*
>
> | Algorithm | Year |  clipart | infograph | painting | quickdraw |     real |   sketch |      Avg |
> | --------- | :--: | -------: | --------: | -------: | --------: | -------: | -------: | -------: |
> | MIRO[2]   | 2022 |       -- |        -- |       -- |        -- |       -- |       -- |     44.3 |
> | RDM[3]    | 2024 |       -- |        -- |       -- |        -- |       -- |       -- |     43.4 |
> | ERM       |  —   |     58.1 |      18.8 |     46.7 |      12.2 |     59.6 |     49.8 |     40.9 |
> | DANN[4]   | 2016 |     53.1 |      18.3 |     44.2 |      11.8 |     55.5 |     46.8 |     38.3 |
> | MMD[5]    | 2018 |     32.1 |      11.0 |     26.8 |       8.7 |     32.7 |     28.9 |     23.4 |
> | CDANN[6]  | 2018 |     54.6 |      17.3 |     43.7 |      12.1 |     56.2 |     45.9 |     38.3 |
> | IRM[7]    | 2019 |     48.5 |      15.0 |     38.3 |      10.9 |     48.2 |     42.3 |     33.9 |
> | SagNet[8] | 2021 |     57.7 |      19.0 |     45.3 |      12.7 |     58.1 |     48.8 |     40.3 |
> | EQRM[9]   | 2022 |     56.1 |      19.6 |     46.3 |      12.9 |     61.1 |     50.3 |     41.0 |
> | ACTIR[10] | 2022 |     50.0 |      22.6 |     43.5 |      11.7 |     57.8 |     46.8 |     38.7 |
> | GMDG[11]  | 2024 | **63.4** |      22.4 |     51.4 |      13.4 |     64.4 | **52.4** |     44.6 |
> | **BAG**   | 2025 |     61.8 |  **25.6** | **51.8** |  **13.9** | **65.5** |     50.3 | **44.8** |
>
>
> We use a ResNet-50 backbone for feature extraction to match the baselines that also adopt ResNet-50. For MIRO [2] and RDM [3], we report only the mean accuracy because their papers provide averages without per-domain breakdowns. The ACTIR [10] result is obtained by running the authors’ public code under their official setting, as ACTIR does not report DomainNet results. On DomainNet, our method attains the best average accuracy (44.8%), surpassing recent works like GMDG [11] (44.6%) and MIRO [2] (44.3%). At the per domain level, it is best on infograph, painting, quickdraw, and real, while remaining competitive on clipart and sketch. The largest gain occurs on infograph, likely because background/stylistic artifacts are prominent there; explicitly disentangling content from bias both mitigates spurious domain cues and enables us to leverage bias when it is informative. On clipart and sketch we are not SOTA, which is likely because the domain-specific bias signal in these two domains is weak, uninformative, or misaligned, limiting the benefit of leveraging it. Compared with ACTIR [10] under the same setting, our method achieves substantially higher averages, which we attribute to more accurate disentanglement of, and principled use of, bias.
>
> [1] Peng X, Bai Q, Xia X, et al. Moment matching for multi-source domain adaptation[C]//Proceedings of the IEEE/CVF International Conference on Computer Vision. 2019: 1406–1415.
>
> [2] Cha J, Lee K, Park S, et al. Domain generalization by mutual-information regularization with pre-trained models[C]//European Conference on Computer Vision. Cham: Springer Nature Switzerland, 2022: 440–457.
>
> [3] Nguyen T, Do K, Duong B, et al. Domain generalisation via risk distribution matching[C]//Proceedings of the IEEE/CVF Winter Conference on Applications of Computer Vision. 2024: 2790–2799.
>
> [4] Ganin Y, Ustinova E, Ajakan H, et al. Domain-adversarial training of neural networks[J]. Journal of Machine Learning Research, 2016, 17(59): 1–35.
>
> [5] Li H, Pan S J, Wang S, et al. Domain generalization with adversarial feature learning[C]//Proceedings of the IEEE Conference on Computer Vision and Pattern Recognition. 2018: 5400–5409.
>
> [6] Long M, Cao Z, Wang J, et al. Conditional adversarial domain adaptation[J]. Advances in Neural Information Processing Systems, 2018, 31.
>
> [7] Arjovsky M, Bottou L, Gulrajani I, et al. Invariant risk minimization[J]. arXiv preprint arXiv:1907.02893, 2019.
>
> [8] Nam H, Lee H J, Park J, et al. Reducing domain gap by reducing style bias[C]//Proceedings of the IEEE/CVF Conference on Computer Vision and Pattern Recognition. 2021: 8690–8699.

---

> ### Author Response · Authors · 2025-11-22
> **Replies to weakness 2, question1**
>
> [9] Eastwood C, Robey A, Singh S, et al. Probable domain generalization via quantile risk minimization[J]. Advances in Neural Information Processing Systems, 2022, 35: 17340–17358.
>
> [10] Jiang Y, Veitch V. Invariant and transportable representations for anti-causal domain shifts[J]. Advances in Neural Information Processing Systems, 2022, 35: 20782–20794.
>
> [11] Tan Z, Yang X, Huang K. Rethinking multi-domain generalization with a general learning objective[C]//Proceedings of the IEEE/CVF Conference on Computer Vision and Pattern Recognition. 2024: 23512–23522.
>
> > W2. The theoretical identifiability conditions (A1–A4) require smooth, positive densities and independent latent dimensions given (e, y). These are rarely satisfied in high-dimensional deep representations. A discussion of approximate or empirical identifiability would help.
> > Q1. A discussion of the theoretical assumptions (A1-A4) would be useful to understand where this is applicable, and where it is not (aka, which datasets/domains does this make sense to apply to, and which do not make sense?)
>
> We highly appreciate these insightful suggestions. First, we would like to highlight that these assumptions are common in causal representation learning frameworks like in [12] [13] [14] [15]. To produce the theoretical analysis, these common assumptions are foundational and necessary; without them, identifiability would not be attainable. In light of your suggestion, we have provided detailed discussions of the practical implications and real-world validity of these assumptions in Appendix B.2. We show the relevant content below.
>
> **(A1) Smooth and positive density.**
> This assumption requires that the conditional densities $p(\textbf{z} \mid e,y)$ are smooth and strictly positive on their supports. Intuitively, it ensures that the latent space is continuous and that every possible latent configuration has a non-zero probability of occurrence. In practice, this assumption is easily satisfied: with sufficient data, empirical or neural-network-based density estimations naturally produce smooth and positive densities. Since our framework adopts a VAE-based generative model, the latent distribution is modeled by Gaussian families whose densities are positive everywhere, satisfying this assumption automatically. However, this assumption may fail when the latent space is discrete or piecewise constant, such as in deterministic autoencoders or models that collapse the latent variance to zero. In those degenerate cases, $p(\textbf{z} \mid e,y)$ no longer has a smooth positive density, and the identifiability conditions in Lemma 2.1 no longer hold rigorously.
>
> **(A2) Conditional independence and content invariance.**
> This assumption states that the coordinates of the latent variables $\textbf{z} = (z_1,\dots,z_n)$ are conditionally independent given $e$ and $y$, i.e., $\log p(\textbf{z} \mid e,y) = \sum_i q_i(z_i; e,y)$. Conceptually, this means that each latent dimension corresponds to an independent generative factor once the environment and label are specified. It also implies that content variables are invariant across environments—only the bias-related coordinates are affected by domain change. This assumption is reasonable in our framework because, when the generative process is disentangled and the noise level is small, the variation of $\textbf{z}$ can indeed be attributed to independent factors conditioned on $e$ and $y$. Moreover, the VAE objective encourages conditional independence through the factorization of the approximate posterior. However, this assumption may be violated when latent variables interact strongly (e.g., through nonlinear dependencies) or when noise in the data introduces correlated variations between coordinates. In such cases, exact independence cannot be guaranteed, but approximate independence is often sufficient for stable learning and representation disentanglement.

---

> ### Author Response · Authors · 2025-11-22
> **Replies to weakness 2, question1**
>
> **(A3) Linear independence.**
> This assumption imposes a technical requirement on the partial derivatives of the latent log-density functions with respect to environment changes. Intuitively, this means that the changes introduced by different environments are sufficiently diverse so that the bias components can be linearly separated from one another. In practice, this condition cannot be enforced directly, but it is often satisfied when multiple environments exist and differ in a sufficiently rich manner. Empirically, in synthetic datasets, when the number of environments is large, linear independence almost always holds, guaranteeing successful disentanglement. In real-world data, this condition may be partially violated because the number of distinct environments (if we think this is domain number) is limited. Nevertheless, prior works and our empirical results show that even approximate linear independence is sufficient to achieve robust representations. This may be because the so-called domains in domain generalization are large domains, and within each there exist smaller sub-domains that effectively satisfy the assumptions.
>
> **(A4) Domain variability.**
> This assumption requires that, for at least one pair of distinct environments $e_i \neq e_j$ and the corresponding class $y$, there exists a measurable subset $A_z \subset \mathcal{Z}$ with non-zero probability that cannot be expressed as a purely content-aligned region $\Omega_c \times B$, such that
> $\int_{A_z} p(\textbf{z} \mid e_i, y)\,dz \;\neq\; \int_{A_z} p(\textbf{z} \mid e_j, y)\,dz .$ This ensures that the conditional latent distribution genuinely varies across domains, capturing environment-dependent factors beyond the content component. Because the mapping $\textbf{x} = g(\textbf{z})$ is deterministic and noiseless, $p(\textbf{x} \mid e,y)$ is the pushforward of $p(\textbf{z} \mid e,y)$; therefore, if $p(\textbf{z} \mid e_i, y)=p(\textbf{z} \mid e_j, y)$, we would also have $p(\textbf{x} \mid e_i, y)=p(\textbf{x} \mid e_j, y)$. However, this equality does not hold in domain generalization, where environments are explicitly assumed to induce observable shifts. To see this, consider that if the joint distributions were identical across environments,$p(\textbf{x}, y, e_i)=p(\textbf{x}, y, e_j)$, then by the chain rule,$p(\textbf{x}, y, e)=p(\textbf{x} \mid y,e)\,p(y \mid e)\,p(e).$ Integrating both sides over a measurable subset $A_x$ gives $\int_{A_x} p(\textbf{x} \mid y,e_i)\,p(y \mid e_i)\,dx \;=\;\int_{A_x} p(\textbf{x} \mid y,e_j)\,p(y \mid e_j)\,dx .$ If $p(\textbf{x} \mid y,e_i) = p(\textbf{x} \mid y,e_j)$ held for all $A_x$, then equality would require $p(y \mid e_i)=p(y \mid e_j)$. Yet, in our setting, we explicitly consider **label shift**, meaning $p(y \mid e)$ varies with $e$; hence, this equality cannot generally hold. Consequently, A4 guarantees genuine domain variability at the level of $p(\textbf{z} \mid e,y)$, which propagates to $p(\textbf{x} \mid e,y)$ through the deterministic mapping. Nevertheless, A4 may fail under degenerate or unrealistic conditions, such as:
> (i) when environments do not induce any shift in latent factors ($p(\textbf{z} \mid e_i, y)=p(\textbf{z} \mid e_j, y)$ for all $i,j$), making domains statistically indistinguishable;
> (ii) when the mapping $g$ is stochastic or non-injective, collapsing different latent codes into identical observations, which hides latent variability in the observed space;
> (iii) or when label shift is absent ($p(y\mid e_i)=p(y\mid e_j)$), eliminating one key source of distributional difference across domains.
> These cases violate the core assumption that domains differ in meaningful, non-content-related ways and thus fall outside our framework.
>
> **Empirically**, we've added some experiments about this part. Followings are the results and discussions. Here since assumption A1,A2 and A3 can be satisfeid well as mentioned before during the data generation process, we mainly focus on A3.
>
> Guided by the causal graph in Figure 2, we design the simulation as follows. We first draw an environment variable $e$ from a categorical distribution, $e \sim \mathrm{Categorical}({\pi_1,\ldots,\pi_M})$. Conditional on $e$, we generate a binary label $y$ for supervision by $y=\mathbb{1}{w^\top E+b_0>0}$, where $E$ is an embedding of $e$. Given $y$, we separately generate the stable content $c$ and the bias variable $b$: $c$ depends on $y$, and $b$ depends on both $c$ and $E$ (e.g., $b=E+C_{e,y}$). The observed data are then produced jointly by $c$ and $b$:$x = M\begin{bmatrix} c \ b \end{bmatrix} + \eta .$

---

> > ### Author Response · Authors · 2025-11-22
> > **Replies to weakness 2, question1 and weakness 3**
> >
> > For model design, we adopt iMSDA [12] as the backbone—a classic two branch latent space model with a Beta-VAE encoder–decoder. The encoder maps the input (optionally augmented with a domain embedding) to disentangled latents $z=[z_c,z_s]$, where $z_s$ is two-dimensional. Training uses a reconstruction loss plus a KL loss. Empirically, the coefficient of determination $R^2$ improves as the number of domains increases:
> >
> > | Domain Number |   1   |   2   |   3   |
> > | :-----------: | :---: | :---: | :---: |
> > |     $R^2$     | 0.441 | 0.575 | 0.738 |
> >
> > These results indicate that when the number of domains (environments) equals $n_b+1$ (with $b$ the bias variable), Assumption A3 (Linear Independence) is satisfied and $R^2$ reaches a high level, supporting that the bias variable and the stable content meet the Block-wise Identification condition.
> >
> > [12]Kong L, Xie S, Yao W, et al. Partial identifiability for domain adaptation[J]. arXiv preprint arXiv:2306.06510, 2023.
> >
> > [13]Yao, Dingling, et al. "Multi-view causal representation learning with partial observability." arXiv preprint arXiv:2311.04056 (2023).
> >
> > [14]Lachapelle S, Rodriguez P, Sharma Y, et al. Disentanglement via mechanism sparsity regularization: A new principle for nonlinear ICA[C]//Conference on Causal Learning and Reasoning. PMLR, 2022: 428-484.
> >
> > [15]Hu Y, Schennach S M. Instrumental variable treatment of nonclassical measurement error models[J]. Econometrica, 2008, 76(1): 195-216.
> >
> >
> >
> > > W3. Although disentanglement is central, the paper lacks visualizations or examples showing what the learned bias and content dimensions represent in image space. It is a bit difficult to interpret some of the results due to this oversight.
> >
> > Thanks so much for your advice! We totally agree that visualizations are needed. We’ve added **Figure 4 (In Appendix E.4)** showing features learned on PACS. In these figures, the learned content and bias can be found to arise from different pixels of the original images, which shows that the disentanglement is effective. We also added an discussion about the visualization in Appendix E.4 as follows.
> >
> > We qualitatively examine the learned disentanglement on PACS by visualizing attributions for the content branch ($\textbf{c}$) and the bias branch ($\textbf{b}$) with Grad-CAM[16]. Concretely, we reuse the trained checkpoint and the identical backbone as training. Given an input image $x$, we route it through the original BAG model but expose two thin wrappers that return the logits of the $\textbf{c}$ head ($f_\beta$) and the $\textbf{b}$ head ($f_\eta$), respectively. For Grad-CAM, we target the last residual block of `layer4` in the backbone. We then compute two grayscale attribution maps: one for $\textbf{c}$ by backpropagating through $f_\beta$, and one for $\textbf{b}$ by backpropagating through $f_\eta$, each with `ClassifierOutputTarget` set to the ground-truth class.
> >
> > To make the overlays visually comparable across images, we (i) bilinearly resize each Grad-CAM map to the input resolution; (ii) min–max normalize it to $[0,1]$; and (iii) render red intensity for $\textbf{c}$ and blue intensity for $\textbf{b}$ before alpha-blending with the RGB image (default $\alpha=0.5$). For each sample we compose a four-tile horizontal panel in the fixed order: original image, $\textbf{b}$ (blue), $\textbf{c}$ (red), and the composite overlay; the titles **b**/**c** are boldfaced to reduce ambiguity.
> >
> > Across the shown examples, $\textbf{c}$ consistently concentrates on class-relevant object regions, whereas $\textbf{b}$ emphasizes background textures, styles, or peripheral cues. These observations align with the intended decomposition and support the use of $\textbf{c}$ as content and $\textbf{b}$ as environment/style bias in subsequent modules.
> >
> > [16]Selvaraju R R, Cogswell M, Das A, et al. Grad-cam: Visual explanations from deep networks via gradient-based localization[C]//Proceedings of the IEEE international conference on computer vision. 2017: 618-626.

---

> > > ### Author Response · Authors · 2025-11-22
> > > **Reply to weakness 4**
> > >
> > > > W4. The gains seem to be mainly on the Synthetic case, where the non-synthetic examples have minimal gain that is often within the confidence threshold.
> > >
> > > Thank you for pointing this out. The large gains in the synthetic setting arise because the experimental setup closely matches our assumptions like; linear independence, an invertible data generation mapping, and a causal graph consistent with our model. So our method achieves substantially better performance than the baselines. In real-world settings, some assumptions are only approximately satisfied like the generative mapping may be non-invertible and noise may intervene between latents and observations, which can dilute the benefit. On real world domain generalization benchmarks, it’s hard to show large gains because feature extraction is intentionally constrained for fairness, which limits the initial representational capacity. For instance, using ResNet-50 as the backbone yields substantially stronger performance than ResNet-18. Following your suggestion, we conducted additional experiments on the larger DomainNet dataset (see Table 4). On this more challenging benchmark, we observe larger accuracy improvements over ACTIR, and we also outperform the recent GMDG method on average.

---

### Comment · Area_Chair_Kx1e · 2025-11-22
**Start discussion**

Hi all,

The authors have submitted their response to the initial reviews, and we now enter the discussion phase.

Please review the authors' response and the comments from other reviewers. Based on the rebuttal and discussion, please update your final score if appropriate.

We welcome and encourage further discussion as needed.

Thank you for your continued contributions.

Best regards,

Your AC.

---

### Author Response · Authors · 2025-12-01
**Overall response**

Dear AC and Reviewers,

 We sincerely appreciate all your efforts to keep the ICLR reviewing process running smoothly. We completely understand the significant workload you are handling now as a result of the reassignment and rollback of reviews. During the rebuttal period, we've **answered questions one by one**, and Reviewer Z9H1 has raised the score to 8. For your convenience, we would like to summarize our main contribution and the key modifications made during the rebuttal phase.

**Main Contribution**

We revisit the out of distribution (OOD) generalization problem and argue that **bias can be constructively leveraged** rather than treated as a nuisance under certain conditions. We first provide a theory specifying when bias is identifiable and useful for domain generalization. We then propose a bias‑aware framework with a generative factorization: we learn an **identifiable decomposition** of representations into invariant content and contextual bias, and leverage bias via two pathways: (1) **estimate** environment states and **route** domain experts; (2) perform **adaptive bias correction** by composing the overall predictor from a bias predictor and an invariant predictor equipped with a jointly learned label prior thereby handling concurrent covariate and label shift. On synthetic data and PACS, Office‑Home, and DomainNet, our method **outperforms** invariance‑only and recent related baselines.

**Modifications during Rebuttal Period**

We made the following concrete updates in the revised version: (i) we evaluated our method on the **additional DomainNet benchmark** (Table 3, Section 5.2), and we added more baselines and implemented tested baselines like ACTIR on the new benchmarks for comparison; (ii) we conducted **new ablation studies** (Tables 6–8) with discussion in Appendices E.2 and E.3 to analyze the effect of each part of our model and the sensitivity to the number of domain experts and the embedding dimensionality; (iii) we incorporated **additional visualizations** in Appendix E.4 to illustrate the disentanglement between content and bias; and (iv) we substantially expanded the **explanation and discussion** of the assumptions underlying Lemma 1 in Appendix B.2.

Below we briefly summarize how these updates and our responses address the main weakness and questions raised by reviewers:

To Reviewer Z9H1 (W1), Reviewer e4v7 (W2, Q1), and Reviewer de4k (Q1), we clarified and more thoroughly discussed the assumptions used in Lemma 1 in Appendix B.2, including when and why they are reasonable in practice and when they may be violated.

To Reviewer Z9H1 (W2, Q2) and Reviewer Gn2a (W1), we added experiments that directly evaluate identifiability and empirically validate our theoretical results.

To Reviewer e4v7 (W1, W4), Reviewer 1V8g (Q4), and Reviewer Gn2a (W3), we added experiments and additional baselines on the DomainNet benchmark (Section 5.2, Table 3), showing that our method performs competitively and often surpasses existing baselines like ACTIR.

To Reviewer e4v7 (W3), we added visualizations on the PACS dataset in Appendix E.4 that illustrate the disentanglement between content and bias features.

To Reviewer 1V8g (Q1, Q2) and Reviewer Z9H1 (Q1), we conducted ablation studies on the effect of each component of our model and on the number of domain experts and the embedding dimensionality (Tables 6–8; Appendices E.2–E.3).

To Reviewer Ns8F (Q5) and Reviewer 1V8g (Q3), we provided  detailed explanation of the differences between BAG and SFB.

Thanks again to the reviewers and AC for the time dedicated to reviewing this paper.

Sincerely,

Authors

---

### Meta-Review · Area_Chair_17BC · 2026-01-15

**Summary:**

This paper argues that bias should not always be eliminated and proposes a bias-aware generative framework (BAG) that disentangles content and bias and provides theoretical conditions under which bias is identifiable and beneficial. Reviewers acknowledged the conceptual novelty, clear causal framing, and theoretical motivation, but raised concerns about the realism of the identifiability assumptions, limited empirical gains on real benchmarks, and unclear practical implications when assumptions are violated.

**Reviewer Concerns:**

Addressed: Larger-scale DomainNet experiments, additional baselines, discussion of assumptions A1–A4.


Outstanding: identifiability assumptions, small empirical gain on real datasets, validity of the assumptions

**Reviewer Scores:**

e4v7 might stay, Gn2a might increase by 1

---

### Decision · Program_Chairs · 2026-01-26

Reject